# *MADS1*-regulated lemma and awn development benefits barley yield

Yueya Zhang [1,7], Chaoqun Shen [1,2,7], Gang Li [2,6] ✉, Jin Shi [1], Yajing Yuan [1], Lingzhen Ye [3], Qingfeng Song [4], Jianxin Shi [1,5] ✉ & Dabing Zhang [1,2,5,8]

Floral organ shape and size in cereal crops can affect grain size and yield, so genes that regulate their development are promising breeding targets. The lemma, which protects inner floral organs, can physically constrain grain growth; while the awn, a needle-like extension of the lemma, creates photosynthate to developing grain. Although several genes and modules controlling grain size and awn/lemma growth in rice have been characterized, these processes, and the relationships between them, are not well understood for barley and wheat. Here, we demonstrate that the barley E-class gene *HvMADS1* positively regulates awn length and lemma width, affecting grain size and weight. Cytological data indicates that *HvMADS1* promotes awn and lemma growth by promoting cell proliferation, while multi-omics data reveals that HvMADS1 target genes are associated with cell cycle, phytohormone signaling, and developmental processes. We define two potential targets of HvMADS1 regulation, *HvSHI* and *HvDL*, whose knockout mutants mimic awn and/or lemma phenotypes of *mads1* mutants. Additionally, we demonstrate that HvMADS1 interacts with APETALA2 (A-class) to synergistically activate downstream genes in awn/lemma development in barley. Notably, we find that *MADS1* function remains conserved in wheat, promoting cell proliferation to increase awn length. These findings extend our understanding of *MADS1* function in floral organ development and provide insights for *Triticeae* crop improvement strategies.

Barley (*Hordeum vulgare*) is one of the most important crops worldwide, mainly used for feed, food, and malt (beer and whisky) production[1]. Crop yield is directly related to grain size and weight[2]. The grain of barley, like that of most *Poaceae* grasses including rice (*Oryza sativa*), wheat (*Triticum aestivum*) and sorghum (*Sorghum bicolor*), is enclosed in spikelet hull (lemma and palea)[3]. In rice, the spikelet hull limits grain growth and thus affects final grain size and yield[4]. A series

of proteins that control grain size and weight by regulating cell proliferation in the spikelet hull has been defined, including the E3 ubiquitin ligase GRAIN WIDTH AND WEIGHT 2 (GW2)[5], the squamosa promoter-binding protein SPL16[6], the calmodulin-binding protein GW5[7], the G-protein γ subunit RGG1[8], the MKKK10−MKK4−MAPK6 cassette[9], the glutaredoxin protein WIDE GRAIN 1 (WG1)[10], and the ubiquitin receptor HOMOLOG OF DA1 ON RICE CHROMOSOME 3

[1]Joint International Research Laboratory of Metabolic and Developmental Sciences, School of Life Sciences and Biotechnology, Shanghai Jiao Tong University, Shanghai 200240, China. [2]School of Agriculture, Food and Wine, The University of Adelaide, Waite Campus, Adelaide, SA 5064, Australia. [3]Department of Agronomy, College of Agriculture and Biotechnology, Zhejiang University, Hangzhou 310058, China. [4]National Key Laboratory of Plant Molecular Genetics, CAS Center for Excellence in Molecular Plant Sciences, Chinese Academy of Sciences, Shanghai 200032, China. [5]Yazhou Bay Institute of Deepsea Sci-Tech, Shanghai Jiao Tong University, Sanya 572025, China. [6]Present address: Department of Plant Pathology, College of Plant Protection, Nanjing Agricultural University, Nanjing 210095, China. [7]These authors contributed equally: Yueya Zhang, Chaoqun Shen. [8]Deceased: Dabing Zhang ✉e-mail: gang.li@njau.edu.cn; jianxin.shi@sjtu.edu.cn

(HDR3)[11]. In barley, however, the relationship between spikelet hull and grain size has not yet been determined, nor the molecular mechanism of grain size regulation defined. It is noteworthy that while the rice lemma and palea together form spikelet shell, the barley lemma entirely encloses the palea such that lemma size may be the most critical factor limiting grain size.

Another floral tissue prevalent in major cereal crops is the awn, a stiff, hair-like structure that extends from the tip of the lemma[12]. The awn is simple and thus often overlooked, but is evolutionarily beneficial for reproduction in wild species[13] where it prevents predation by birds and animals, and aids seed dispersal and self-planting[14–16]. In rice and sorghum, *awnless* is considered a key domestication trait to facilitate seed harvest and storage without affecting yield[13,17,18]. In contrast, the barley and wheat awn has been largely preserved, although domesticated cultivars have shorter awns than their wild ancestors[19], as barley and wheat awns contribute to grain filling through photosynthesis[20–22], especially when the leaves are prematurely senescent or damaged by diseases[23,24]. Genes regulating awn development have been reported in different cereals. In sorghum, duplication of the ALOG transcription factor *awn1* prevents awn development, leading to awn loss during domestication[13,25]. In rice, some genes have been artificially selected during domestication: *Awn-1* (*An-1*) encodes a helix-loop-helix protein and positively controls cell division and the formation of awn primordia[18]; *An-2/LONG AND BARBED AWN* (*LABA1*) encodes a cytokinin-activating enzyme and promotes awn elongation by increasing cytokinin concentration[14,26]; *REGULATOR OF AWN ELONGATION 2* (*RAE2*)/*GRAIN NUMBER, GRAIN LENGTH AND AWN DEVELOPMENT 1* (*GAD1*)[27,28] and *EPIDERMAL PATTERING FACTOR-LIKE 2* (*EPFL2*)[29] encode secreted signal proteins involved in awn development. *DROOPING LEAF* (*DL*, a *YABBY* gene) and *OsETTIN2* (an auxin response factor gene) work together to promote awn formation[30]. Other reported genes include *REGULATOR of AWN ELONGATION 3* encoding an E3 ubiquitin ligase[31], *TONGARI-BOUSHI1* encoding a YABBY protein[32], *CW-ZF7* encoding a CW-domain containing zinc finger protein[33], and *GRAIN LENGTH AND AWN 1* encoding a mitogen-activated protein kinase phosphatase[34]. Less is known about awn development in barley and wheat. In wheat, *FRIZZY PANICLE* promotes awn growth[35], while a zinc finger protein expressed at the *B1* locus suppresses awn development[21,36]. In barley, *SHORT INTERNODES* (*SHI*) controls awn elongation and pistil morphology[22] and *APETALA2* (*AP2*) regulates awn development, awn-lemma boundary and lemma identity[37]. Nevertheless, molecular and genetic mechanisms regulating barley and wheat awn development, as well as relationships between awn and grain yield, remain to be further elucidated.

MADS-box proteins, singly and in combinations, are the main regulators of floral organ development, as summarized in the ABCDE models of *Arabidopsis* and rice[38]. This model classifies their encoding genes into five different categories (A, B, C, D, and E) based on their homeotic functions; class E genes, encoding proteins that interact with the other four classes to regulate the development of each floral organ[39], also known as *SEPALLATA* (*SEP*) genes that participate in a variety of other developmental processes, including spike architecture, flowering time, fruit ripening, and inflorescence thermomorphogenesis[40]. Rice and barley contain 5 *SEP* genes (*MADS1*, *MADS5*, *MADS34*, *MADS7* and *MADS8*), but a significant expansion has occurred in wheat[38]. Due to the lack of corresponding floral organ mutants and only a few known floral genes, the ABCDE model in barley and wheat has not been systematically described, and neither has the role of *SEP* genes in reproductive development, especially in floral organ development.

Here, we report that a *SEP* gene, *HvMADS1*, controls barley grain size and weight by regulating awn elongation and lemma transverse growth. We found that *HvMADS1* regulates *HvSHI* and *HvDL*

expression by directly binding to their promoters. The *shi* and *dl* mutations inhibited cell proliferation, creating shorter awns and/or narrower lemmas than wild-type (WT), mimicking the *mads1* phenotype. Moreover, we found that A-class gene encoded HvAP2 interacts with HvMADS1 to promote transcriptional activity, thereby involving in awn and lemma development. We also verified that the role of *MADS1* in controlling awn development is conserved in wheat. Our study provides information to broaden our understanding of barley and wheat floral organ development, and reveals a molecular and genetic role for *HvMADS1* in controlling grain size in barley.

## Results

### *HvMADS1* positively regulates awn development by promoting cell proliferation and further influences spike photosynthesis

Our previous study demonstrated that *HvMADS1* in barley plays an important role in response to high temperature by maintaining a normal, unbranched inflorescence morphology[40]. Notably, as compared with WT, barley *mads1* mutant exhibits shorter awn at ambient temperatures, indicating that *HvMADS1* also regulates awn development[40]; however, the underlying explicit cellular and molecular mechanisms remain unclear.

We generated three knockout lines of *HvMADS1* using CRISPR-Cas9 system, namely *mads1-3*, *mads1-5* and *mads1-8* in barley cv. Golden Promise (Supplementary Fig. 1a), with different alleles from those reported in previous studies[40]. All three homozygous mutant plants also exhibited significantly shorter awns than WT plants (Fig. 1a, d), confirming an important role of *HvMADS1* in awn development as previously reported[40]. Cytological analyses on the central parts of the awns revealed that the width and thickness of *mads1* awns were also significantly smaller than WT (Fig. 1b, d). The cell width in the transverse section of *mads1* awns was slightly smaller than WT, while longitudinal cell length of *mads1* was the same as WT (Fig. 1e); remarkably, cell numbers of *mads1* awns in both directions were substantially reduced as compared with those in WT (Fig. 1e).

As the *mads1* awn was significantly smaller than WT, the corresponding area of chlorenchyma tissue, which contributes to spike photosynthesis and thus grain filling[20], was also obviously reduced (Fig. 1b). Consequently, the whole-spike net photosynthetic rate (spike $A_{net}$) measured on intact and de-awned WT and *mads1* spikes showed that the $A_{net}$ of *mads1* spikes was significantly lower than that of the WT, although slightly higher than that of de-awned *mads1* and WT spikes at a photon flux density (PFD) $\geq 250$ μmol (photons) m$^{-2}$ s$^{-1}$ (Fig. 1c). Most interestingly, the grain weight of de-awned WT was about 6% lower than that of the non-de-awned WT (Supplementary Fig. 1b). Collectively, these results indicated that *HvMADS1* controls cell proliferation to positively regulate awn size, which directly affects spike photosynthesis in barley.

### *HvMADS1* controls lemma width by promoting cell proliferation

*MADS1* in rice is well-known for its multiple roles in floral organ identity specification; the *osmads1* null mutant displays defective inner floral organs, and an elongated leafy lemma and palea[39,41]. The spikelets of barley *mads1* mutant contained a pair of glumes, a lemma, a palea, and three whorls of internal floral organs, including two lodicules, three stamens, and a carpel, all of which morphologically resembled WT organs (Supplementary Fig. 1c, d, e). However, unlike WT, the *mads1* lemma did not completely envelop the palea (Supplementary Fig. 1c, e). Further phenotypic observations found that lemma width, but not palea width or spikelet hull length, is significantly reduced in *mads1* spikelets (Fig. 1f, g, i and Supplementary Fig. 1f, g). The width of inner epidermal cell of *mads1* was indistinguishable from that of WT (Fig. 1g–i), as were the outer parenchyma cell width (Fig. 1j–l). Cytological analyses revealed that it is the cell number across the lemma

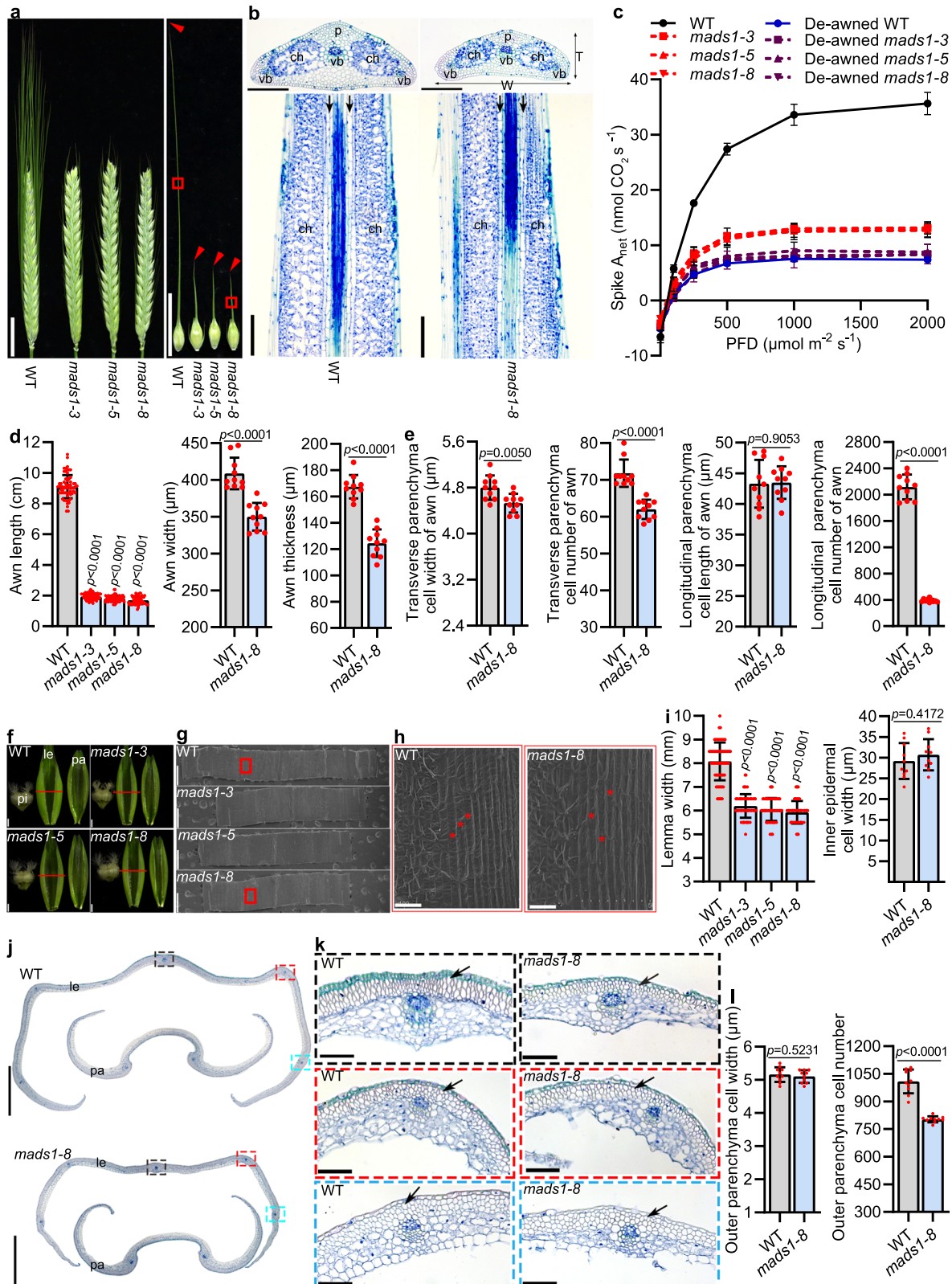

width that is reduced in *mads1* plants (Fig. 1l). Thus, *HvMADS1* also positively regulates lemma width by promoting cell proliferation.

### The *mads1* mutant produces small grains

In barley and wheat, awns provide photosynthate for developing grains[20,21]. To examine whether *HvMADS1* mediated awn size and lemma size ultimately affects grain size, we compared grain

characteristics of WT and *mads1* plants along grain development process. The obvious differences in grain size and weight were observed from 16 days after fertilization (DAF), in which the weight of *mads1* grain was significantly lower than those of WT (Fig. 2a, b). Although there was a slight increase in length, significant decreases in both width and thickness of *mads1* grain were observed as compared to WT (Fig. 2c–e and Supplementary Fig. 2a, b), which was consistent

**Fig. 1 | *HvMADS1* regulates lemma and awn development through controlling cell proliferation, which in turn influences spike photosynthesis. a** Images of spike and awn of WT and *mads1* knockout lines. Red arrowheads show awn tips; **b** Transverse (upper) and longitudinal (lower) sections of red boxed regions in (**a**). Black arrows indicate cells used for cell length measurement and cell number estimation in (**e**). **c** Photosynthetic light response curves of intact and de-awned WT and *mads1* spikes (*n* = 3 biological replicates). PFD, photon flux density; Spike A$_{net}$, whole-spike net photosynthetic rate. **d** Statistic data of length, width, and thickness of WT and *mads1* awns. **e** Statistic data of cell length and cell number in both transverse and longitudinal sections of WT and *mads1* awns (*n* = 10 biologically independent samples). **f** Images of spikelet hulls of WT and *mads1*. Red lines indicate the position of the cross-section and location of the area used for scanning

electron microscopy (SEM) in (**g**) and (**h**). **g** SEM images of the inner surface of WT and *mads1* lemmas. **h** Magnified view of the red boxed lemma area in (**g**). **i** Statistic data of lemma width and inner epidermal cell width of WT and *mads1* lemmas. **j** A cross-section of WT and *mads1* spikelet hull. **k** Magnified images of the boxed lemma in (**j**). Black arrows indicate outer parenchymal cell layers. **l** Statistic data of the width and number of outer parenchyma cell across WT and *mads1* lemmas (*n* = 10 biologically independent samples). Values are mean ± SD, *p* values obtained from two-tailed Student's t-test; Scale bars, 2 cm (**a**), 100 μm (**b, h**), 1 mm (**f**), 0.8 mm (**g**), 500 μm (**j**), 50 μm (**k**). ch chlorenchyma tissue, le lemma, p parenchyma cells, pa palea, pi pistil, T thickness, vb vascular bundle, W width. Source data are provided as a Source Data file.

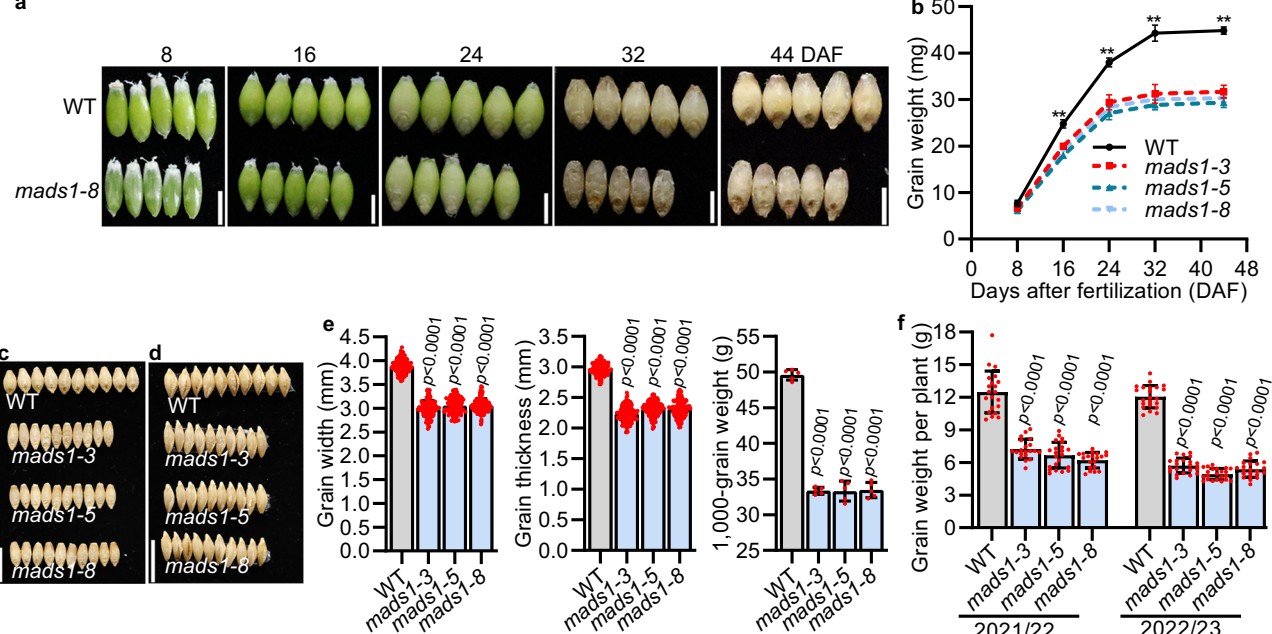

**Fig. 2 | *HvMADS1* influences grain filling and grain size.** Images (**a**) and staistic data of weight (**b**) of WT and *mads1* grains at different developmental stage (*n* = 3 biological replicates). Images (**c, d**) and statistic data (**e**) of the width, thickness and 1,000-grain weight of mature WT and *mads1* grains (*n* = 150 individual grains for grain size and *n* = 5 biological replicates for grain weight). **f** Staistic data of grain

weight per plant of WT and *mads1* grown in the paddy field in Shanghai for two consecutive years (2021–22 and 2022–23). Values are mean ± SD, two-tailed Student's t-test was used to generate the *p* values; **\*\**p* < 0.01; Scale bars, 0.5 cm (**a**), 1 cm (**c, d**). Source data are provided as a Source Data file.

with observed narrower lemma in *mads1* (Fig. 1f, i). To rule out the effect of carpel size on grain size, we also compared carpels size between WT and *mads1* mutant, and found that the carpel size of the mutant was not significantly smaller than that of the WT (Supplementary Fig. 3a, b). In addition, to rule out the effect of endosperm on grain size we crossed *mads1* (♀) with WT (♂), and found that grain size of all F2 progenies is similar to that of WT (Supplementary Fig. 3c, d). These results implied that the spikelet hull in barley is an important factor limiting grain size as it is in rice, as lemma width can directly affect the grain width and thickness (Fig. 2c–e).

The weight of *mads1* grains was about 33 % lower than that of WT grain (Fig. 2e). This agronomically important result was confirmed in two-year field trials (Supplementary Fig. 2c). It is worth noting that although the spike length, spikelet number per spike and plant of *mads1* were not significantly reduced compared with those of the WT (Supplementary Fig. 2d, e, f), grain numbers per spike and grain numbers per plant of *mads1* were smaller than those of WT (Supplementary Fig. 2g, h). This was mainly due to much lower seed setting rate of *mads1*, which was 32-39% lower than that of WT (Supplementary Fig. 2i). Consequently, the overall grain yield (weight) per plant in *mads1* was decreased 42–59% as compared with that of WT

(Fig. 2f), suggesting that *HvMADS1* is important for sustaining barley grain yield.

## *HvMADS1* expression promotes awn development and lemma transverse growth

To determine the specific stages at which *HvMADS1* affects awn and lemma differentiation and development, we compared WT and *mads1* spikelet development using SEM at different Waddington stages (W2.5–W5.5)[42], including W3.0 and W4.5 when lemma and awn primordia initiate, respectively. At stage W3.0, lemma primordia initiated normally in both WT and *mads1* (Fig. 3a). During stages W4.5–W5.5, awn primordia emerged at the lemma apex similarly in both WT and *mads1*, which, however, elongated rapidly in WT but slowly in *mads1* (Fig. 3a). Afterwards, awn length, lemma width, and the cell number of the lemma increased rapidly in WT but slower in *mads1*, especially after stage W7.5 (Fig. 3b).

To confirm the correlation between *HvMADS1* expression and observed phenotype, we performed gene expression analysis. Quantitative reverse transcription PCR (RT-qPCR) results revealed that *HvMADS1* is relatively high expressed in developing inflorescences at stages W5.0–W9.5, the period of floral organ development

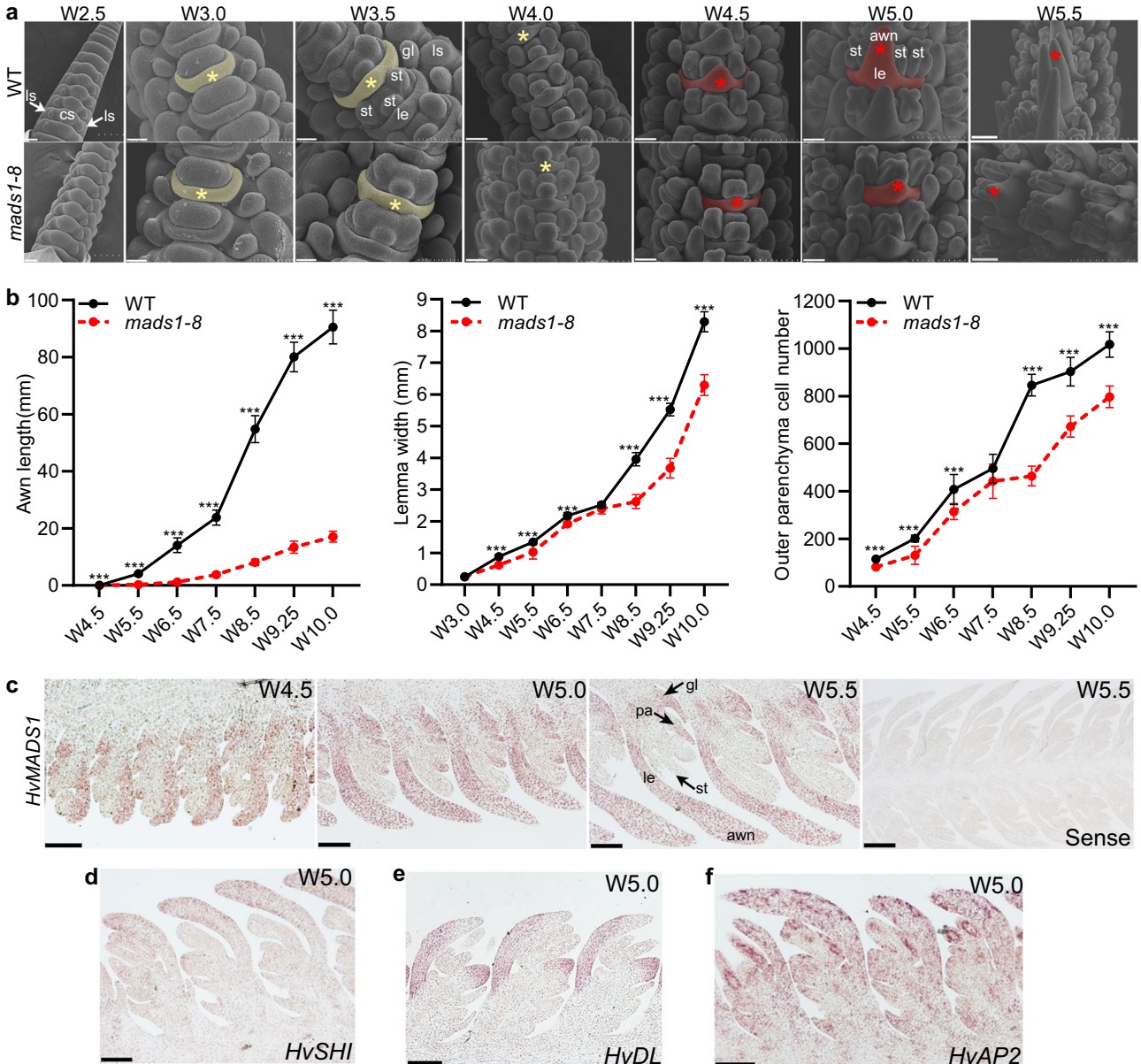

**Fig. 3 | *HvMADS1* spatiotemporally affects lemma and awn development. a** SEM images of WT and *mads1* spikelets at different developmental stages (W2.5–W5.5). Yellow shading and star indicate lemma; red shading and star indicate lemma and awn. Scale bars, 50 μm (W3.0, W3.5), 80 μm (W2.5, W4.0, W4.5, W5.0), 160 μm (W5.5). **b** Dynamic changes of awn length, lemma width, and cell number across the lemma along the process of inflorescence development. The width and cell number of the lemma were obtained by analyzing lemma sections (*n* = 10 individual lemma samples for lemma width at stages W6.5-W10.0). **c–f** Results of RNA in situ hybridization of *HvMADS1* (**c**), *SHORT INTERNODES* (*HvSHI*, **d**), *DROOPING LEAF* (*HvDL*, **e**) and *APETALA2* (*HvAP2*, **f**) in longitudinal sections of developing WT inflorescences. Scale bars, 100 μm (except for W5.5 sense probe panel bar, 250 μm). (**a**, **c–f**) The representative results from three independent replicates were presented. Values are mean ± SD, asterisks indicate statistically significant differences (two-tailed Student's t-tests; ***P < 0.001). cs central spikelet, gl glume, le lemma, ls lateral spikelet, pa palea, pi pistil, st stamen. Source data are provided as a Source Data file.

including rapid growth of the lemma and awn (Supplementary Fig. 4a). RNA in situ hybridization revealed that *HvMADS1* transcripts are abundant in the awn and lemma primordia at stages W4.5, W5.0, and W5.5 (Fig. 3c). Together, these results suggest that specific expression of *HvMADS1* induces rapid cell division/proliferation in the lemma and awn.

### *HvMADS1* regulates hormone- and cell cycle–related genes
To better investigate biological processes influenced by *HvMADS1* mutations, we performed transcriptomic (RNA-seq) analysis of WT and *mads1* lemma and awn at stage W7.5 before the awn longitudinal growth and lemma transverse growth started to differ dramatically

(Fig. 3b). Differential gene expression analysis detected 1,374 up-regulated and 2,872 down-regulated genes in *mads1* relative to WT (Fig. 4a). Gene ontology (GO) term analysis revealed that differentially expressed genes (DEGs) are enriched for biological processes including flower morphogenesis, plant organ development, hormone meta-bolism, response processes, and cell wall development (Fig. 4b). Specifically, some of these DEGs are annotated to be involved in auxin, gibberellin, jasmonic acid, and cytokinin metabolic and signaling processes (Supplementary Fig. 5a), which is consistent with previous reports that there are many hormone-related genes downstream of *MADS1*, and that these four hormone families are involved in barley and/or rice spikelet development[40,43–45]. Additionally, expression of cell

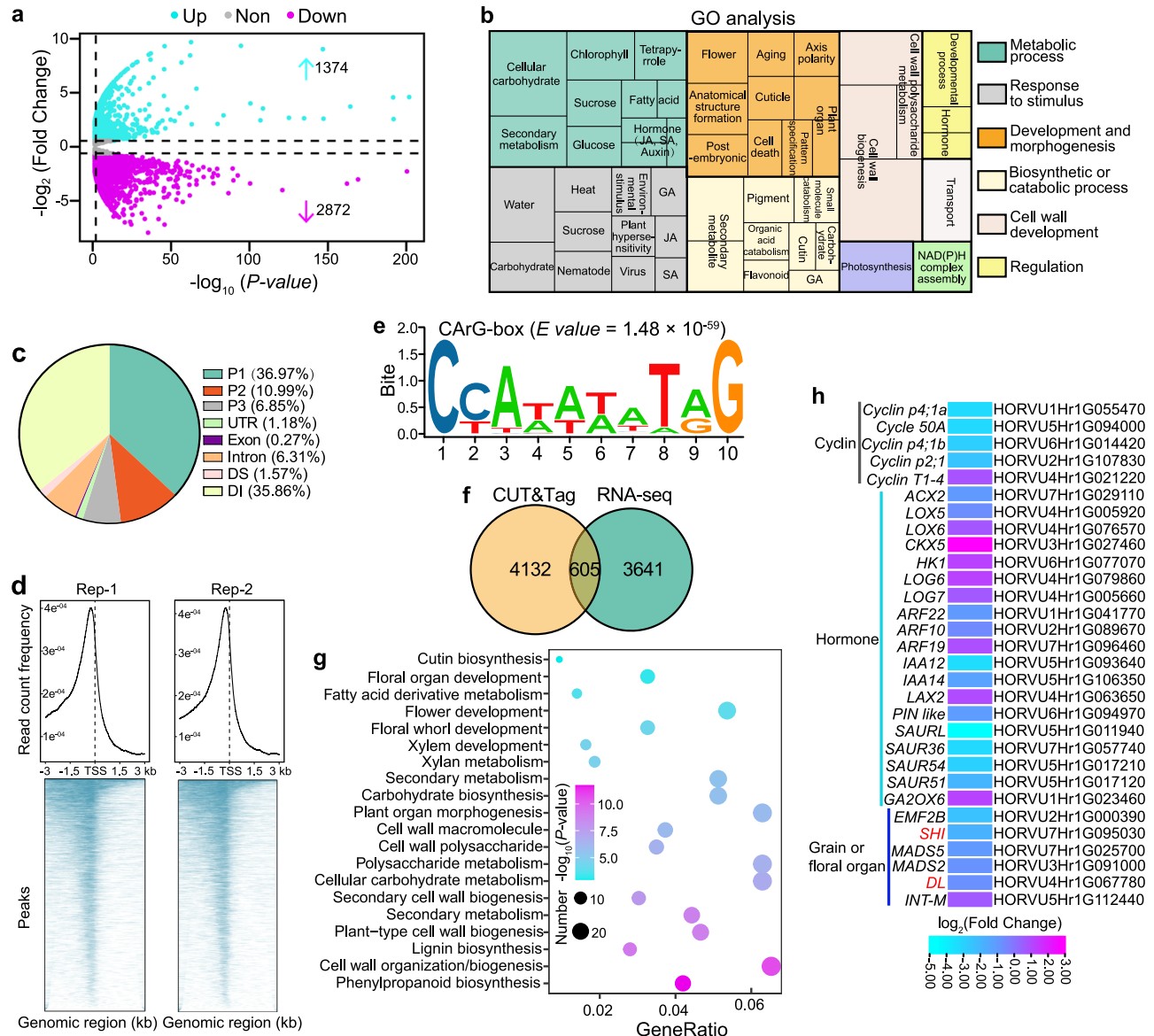

**Fig. 4 | Target genes of MADS1 identified by transcriptome analysis and CUT&Tag. a** The Volcano plot of DEGs in lemma and awns between *mads1* and WT. Blue and purple points represent up- and down-regulated genes, respectively; Gray points represent insignificantly changed genes. Cut-off value was set to false discovery rate (FDR) < 0.05, absolute fold change ≥ 1.5, FDR was adjusted by Benjamini–Hochberg correction. **b** Gene Ontology (GO) analysis of DEGs in (**a**). **c** CUT&Tag analysis of MADS1-binding regions in genomic regions of direct target genes: P1, 0–1 kb upstream of start codon; P2, 1–2 kb upstream of start codon; P3, 2–3 kb upstream of start codon; DS, 1–300 bp downstream of stop codon; DI, >300 bp downstream of stop codon. **d** The distribution of CUT&Tag peaks around transcriptional start sites (TSS). **e** Preferred CArG-box sequence for HvMADS1 binding. **f** The Venn diagram showing overlapped genes jointly identified by RNA-seq and CUT&Tag analyses. **g** GO enrichment analysis of 605 putative direct target genes identified from (**f**), *p* values were calculated by the Fisher method. **h** A heatmap showing expression patterns of putative HvMADS1 direct target genes in lemmas and awns between *mads1* and WT.

cycle-related genes was generally down-regulated in *mads1* lemmas and awns, which was further confirmed by RT-qPCR (Supplementary Fig. 5b, c), consisting with the finding that *HvMADS1* promotes cell proliferation in these tissues (Fig. 1).

To further unearth direct targets of HvMADS1 regulation, we performed Cleavage Under Targets and Tagmentation (CUT&Tag) assays using stage W7.5 lemma and awn from an existing *HvMADS1-eGFP* transgenic line[40]. We identified 4,737 predicted direct target genes (Supplementary Data 1); approximately half of HvMADS1 binding sites for these genes were located in the 2 kb promoter region upstream of the start codon (Fig. 4c), in agreement with the idea that HvMADS1 acts as a transcription factor. The highest frequency of binding occurred around the transcriptional start site (TSS; Fig. 4d), while one of the most enriched sequences for the in vivo HvMADS1-

binding motif (E value = 1.48 × 10⁻⁵⁹) was a *cis*-element CArG-box (Fig. 4e).

Intersection of RNA-seq and CUT&Tag data identified 605 genes as the most likely direct targets of HvMADS1 (Fig. 4f and Supplementary Data 2). Further GO analysis revealed that these targets are also significantly enriched in floral organ development, plant organ morphogenesis, and cell wall-related processes (Fig. 4g). Notably, the heatmap and genome browser view showed that *HvSHI* and *HvDL* expression is significantly suppressed in *mads1* lemmas and awns (Figs. 4h, 5a, j and Supplementary Data 3). *HvSHI* is considered as a strong candidate for the *short awn 2* (*Lks2*) trait involved in barley awn elongation[22], whereas homologs of *HvDL* is involved in awn development in wheat and rice[30,35]. Our combined methods also revealed several other direct candidate targets of HvMADS1 that are involved in

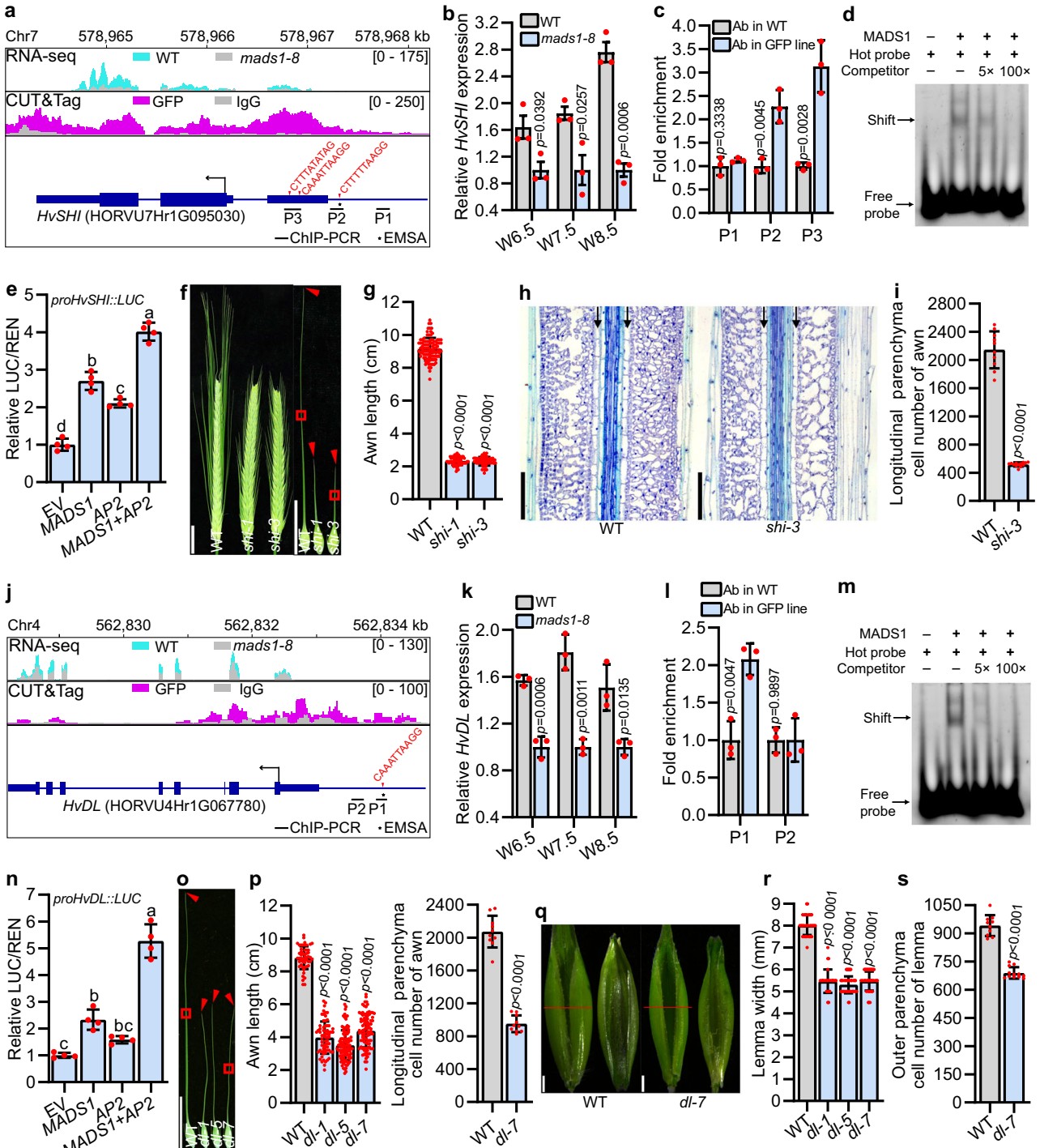

**Fig. 5 | HvMADS1 directly regulates potential downstream genes *HvSHI* and *HvDL* to control awn length and lemma width. a**, **j** Genome browser view of RNA-seq and CUT&Tag profiles and the schematic diagram of promoter fragments of *HvSHI* (**a**) and *HvDL* (**j**) for ChIP-PCR and EMSA assay in (**c**, **d**, **l**, **m**). Expression of *HvSHI* (**b**) and *HvDL* (**k**) in developing lemma and awn of WT and *mads1*. **c**, **l** Binding of HvMADS1 to the *HvSHI* (**c**) and *HvDL* (**l**) promoters in vivo as revealed by ChIP-PCR. Binding of MADS1 to probes from the *HvSHI* (**d**) and *HvDL* (**m**) promoters harboring a CArG-box (EMSA), a representative result from three replicates was presented. Transactivation activity assay of HvMADS1 and HvMADS1-HvAP2 interaction on *HvSHI* (**e**) and *HvDL* (**n**). Different letters represent significant differences (*p* < 0.05) determined by one-way ANOVA with Tukey's multiple comparisons test.

Images of *shi* (**f**) and *dl* (**o**) awns. Longitudinal sections of boxed regions in (**o**) are shown in Supplementary Fig. 7c. **g** Statistic data of WT and *shi* awns. **h** Longitudinal sections in the boxed regions of (**f**). Black arrows indicate cells used to measure cell length for cell number estimation in (**i**). **i**, Statistic data of longitudinal parenchyma cell number in WT and *shi* awns. **p** Statistic data of the length and cell number of WT and *dl* awns. **q** Images of spikelet hulls of WT and *dl*. Red lines indicate the position of the cross-section used in (**r**, **s**). Statistic data of the width (**r**) and number of the outer parenchyma cell (**s**) in the lemma of WT and *dl*. Values are mean ± SD, *p* values shown are from two-tailed Student's t-test; *n* = 3 and 4 biological replicates for (**b**, **c**, **k**, **l**) and (**e**, **n**), respectively. Scale bars, 2 cm (**f**, **o**), 1 mm (**q**), 100 μm (**h**). Source data are provided as a Source Data file.

floral organ development, cell cycle, and auxin response processes (Fig. 4h and Supplementary Fig. 5d).

### HvMADS1 could regulate awn and lemma development through direct regulation of potential targets *HvSHI* and *HvDL*

Expression analysis revealed that both *HvSHI* and *HvDL* are expressed in the developing inflorescence of WT (Supplementary Figs. 6a, 7a), and that their expression levels are significantly downregulated in *mads1* lemma and awn at stages W6.5, W7.5, and W8.5 (Fig. 5b, k). In situ hybridization confirmed the specific expression of *HvSHI* in awn (Fig. 3d and Supplementary Fig. 4b) and *HvDL* in both lemma and awn (Fig. 3e and Supplementary Fig. 4c), suggesting that these two genes likely function in awn and/or lemma development. ChIP-qPCR assays further corroborated that HvMADS1 directly binds in vivo to *HvSHI* and *HvDL* promoter fragments containing a CArG-box motif (Fig. 5c, l), which was subsequently verified in vitro by electrophoretic mobility shift assay (EMSA; Fig. 5d, m). Dual-luciferase (LUC) assays in barley protoplasts showed that HvMADS1 can transiently activate the expression of *HvSHI* and *HvDL* (Fig. 5e, n), confirming that HvMADS1 can directly bind to promoters of *HvSHI* and *HvDL* to activate their transcription.

To confirm whether *HvSHI* and *HvDL* affect barley awn and lemma development, we created *HvSHI* mutants (named *shi-1* and *shi-3*; Supplementary Fig. 6b) and *HvDL* mutants (*dl-1*, *dl-5* and *dl-7*; Supplementary Fig. 7b) via CRISPR/Cas9-mediated gene editing. As in *mads1* mutants, awns of the *shi* and *dl* mutants were significantly shorter than that of WT, primarily due to a reduction in cell number rather than in cell size (Fig. 5f–i, o, p, and Supplementary Fig. 6c, 7c, d). Correspondingly, the 1,000-grain weight of *shi* mutant was significantly smaller than that of WT (Supplementary Fig. 6d).

Notably, our observations showed that the *dl* mutant exhibits a significantly narrower lemma (Fig. 5q, r) but similar spikelet hull length and palea width, as compared with WT (Supplementary Fig. 7e, f). As *dl* mutants were unable to set seeds due to defective pistils with ectopic carpel-like organs (Supplementary Fig. 7g), the grain size of *dl* mutants could not be obtained. Microscopic observation of lemma cross-sections further revealed significantly reduced cell number in *dl*, which mimicked the defect in *mads1* (Fig. 5s and Supplementary Fig. 7h). SEM results confirmed that, similar to *mads1*, it is not the initiation of *dl* lemma but the growth of *shi* and *dl* awns is slowed down as compared to that in WT (Supplementary Fig. 8). Correspondingly, expression levels of a number of cell cycle genes, whose expression was significantly decreased in *mads1*, were also remarkably down-regulated in *shi* and *dl* mutants (Supplementary Fig. 6e, 7i). These results implied that the direct regulation of potential targets *HvSHI* and *HvDL* by HvMADS1 could contribute to HvMADS1-dependent regulation of awn and/or lemma development through controlling cell proliferation.

### Interaction of HvMADS1 with HvAP2

SEP proteins, to which MADS1 belongs, often form higher-order complexes with other ABCDE proteins to regulate floral organ development[46,47]. Our yeast two-hybrid (Y2H) assay confirmed above mentioned idea and found that HvMADS1 interacts with other A-, B-, C-, D-, E-class, and AGL6 subfamily homeotic proteins except HvMADS4 and HvMADS14 (Supplementary Fig. 9). Most interestingly, HvMADS1 interacted with HvAP2 in yeast cells (Fig. 6a). Considering the fact that HvAP2 was reported to regulate awn development[37], we speculate that HvMADS1 and HvAP2 might function together in barley awn development. The interaction of the HvAP2 with HvMADS1 was subsequently confirmed by split firefly luciferase complementation (SFLC) assays and bimolecular florescence complementation (BiFC) assays in tobacco epidermal cells (Fig. 6b, c), and further validated by in vitro pull down and in vivo Co-immunoprecipitation (Co-IP) assays (Fig. 6d, e). Further investigations showed that the intervening (I) and keratin-like (K) domain of HvMADS1 are required for physical interaction with

HvAP2 (Supplementary Fig. 10a), and that HvAP2 interacts with HvMADS1 likely through its C- and N-terminal domain (Supplementary Fig. 10b). A transient transcriptional activity assay demonstrated that the interaction of HvMADS1 with HvAP2 significantly enhances transcriptional activity of HvMADS1 (Fig. 6f). Additional RT-qPCR and in situ hybridization revealed that *HvAP2* is ubiquitously expressed in developing floral organs, including in the awn and lemma during inflorescence development (Fig. 3f, Supplementary Figs. 4d and 11a), which overlapped with that of *HvMADS1*.

To further explore the functions of HvAP2, we generated *ap2* knockout mutants (*ap2-8* and *ap2-11*; Supplementary Fig. 11b). Phenotypic analysis confirmed that *HvAP2* also promotes awn growth mainly by affecting cell number (Fig. 6g, h, and Supplementary Fig. 8, 11c, d), and that the *ap2* lemma is also slightly narrower due to the reduced cell number (Fig. 6i–k). The width and thickness of *ap2* grain were significantly narrower than WT (Supplementary Fig. 11g, h). The observed phenotypes of awn length and grain size of *ap2* mutants are consistent with previous reports[37]. To further test the genetic relationship between MADS1 and AP2 with respect to awn length and lemma width, we created *mads1-8 ap2-11* double mutant by crossing *mads1-8* with *ap2-11*. Phenotypic similarities of the degree of awn length shortening between *mads1-8* and *mads1-8 ap2-11* (Fig. 6l) indicated that *HvMADS1* is genetically epistatic to *HvAP2* in the control of awn growth. The lemma width of double mutant was significantly reduced, much narrower than that of *mads1-8*, *ap2-11* and WT (Fig. 6m). Taken together, these findings indicate that HvMADS1 and HvAP2 synergistically regulate the awn and lemma development. Expression analysis found that expression levels of *HvSHI* and *HvDL*, as well as several cell cycle genes that are downregulated in *mads1*, are also significantly reduced in *ap2* mutants (Supplementary Fig. 11i). Dual-LUC assays demonstrated that *HvSHI* and *HvDL* promoter activation is much significantly enhanced when HvAP2 and HvMADS1 are co-transformed than HvMADS1 alone (Fig. 5e, n). Thus, HvMADS1 and HvAP2 may form a complex and regulate the common target genes *HvSHI* and *HvDL* to control awn size and/or lemma transverse growth.

### The function of bread wheat *MADS1* in regulating awn size

There are three wheat homologs of *HvMADS1*, namely *TaMADS1-A* (TraesCS4A01G058900), *TaMADS1-B* (TraesCS4B01G245700), and *TaMADS1-D* (TraesCS4D01G243700; Supplementary Fig. 12a). To investigate whether *TaMADS1* is involved in wheat awn and lemma development, we generated a triple recessive mutant of *TaMADS1* (*Tam1-abd*) in hexaploid bread wheat cultivar Fielder by CRISPR-Cas9 (Supplementary Fig. 12b). Interestingly, the awns of both central and top spikelets in *Tam1-abd* were significantly smaller compared to those of WT (Fig. 7a–e, g, h). Further histological experiments demonstrated that the number of longitudinal parenchyma cell of the *Tam1-abd* awn was reduced by 31.8 % compared to that of WT, while the cell length was not significantly changed (Fig. 7f, i and Supplementary Fig. 12c). Thus, consistent with barley results, reduction in cell number due to the *Tam1-abd* mutation likely resulted in shorter awns in wheat mutant. Notably, *Tam1-abd* did not show any phenotypes in lemma and other floral organs, due likely to the functional redundancy since *TaMADS1* has duplicated in wheat (Supplementary Fig. 12a).

## Discussion

Grain size is both a critical yield trait and a complex developmental trait regulated by many factors, including the photosynthetic rate of source tissue and spikelet hull size[4,17]. Research on regulators of barley grain size has lagged significantly behind that of rice due to the lack of mutants, artificial populations for genetic analysis, and natural populations with high-quality sequencing data. With the help of CRISPR-Cas9 genome editing technology, we have functionally characterized that *HvMADS1* controls barley grain size and weight by regulating awn length that influences spike photosynthesis and lemma width that

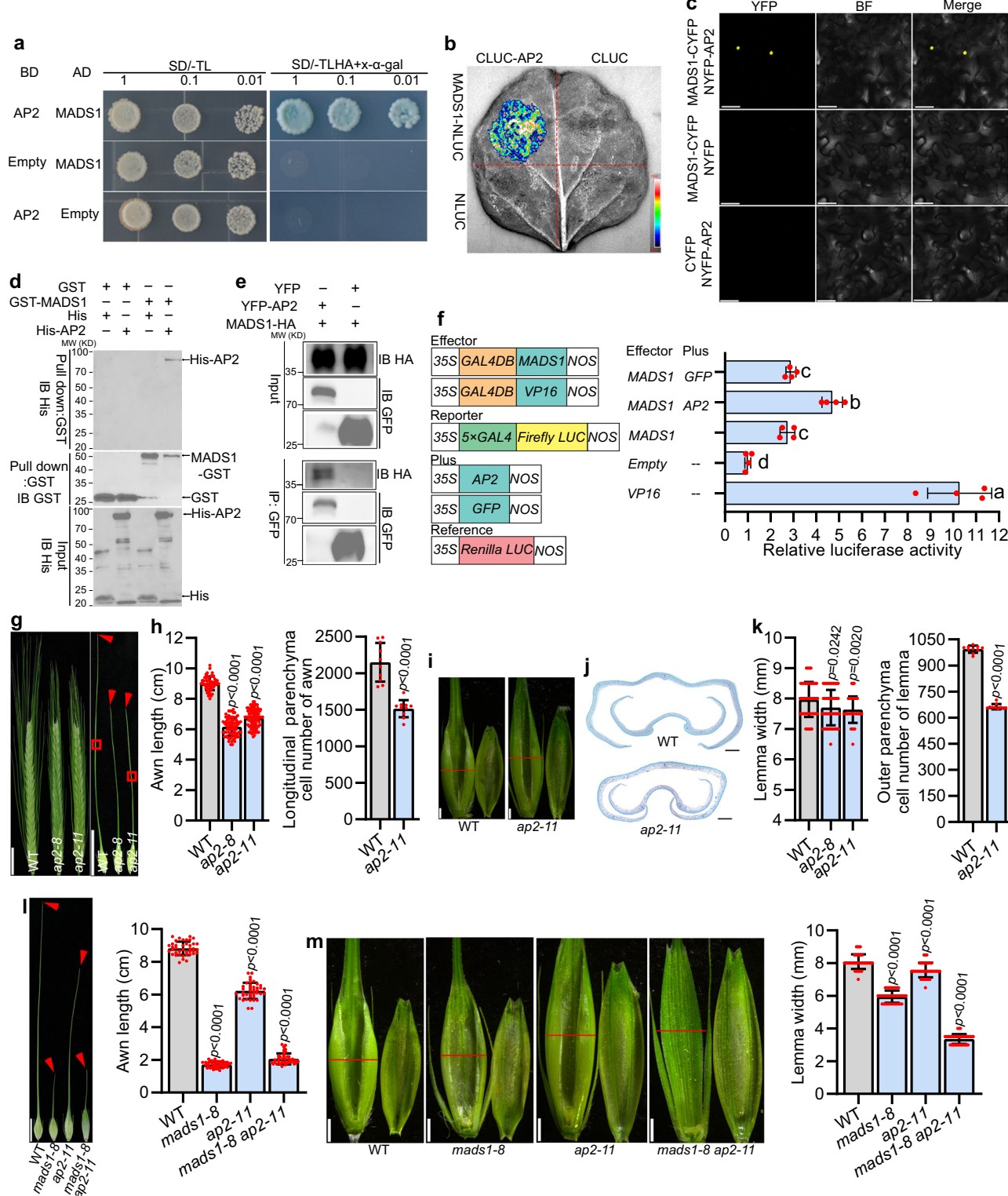

affects spikelet hull size. We have further shown that HvMADS1 directly regulates downstream target genes, and can physically interact with other proteins to do so; its molecular network in barley has been proposed based on our multiple level experimental evidence (Fig. 7j). Notably, the function of *MADS1* in controlling awn length is conserved in wheat. Since *TaDL* has also been reported to be involved in wheat awn development[35], further studies on the regulatory relationship between TaMADS1 and *TaDL* will help to establish the wheat molecular network for awn development.

Divergence in biological function of genes can arise through sub- or neo-functionalization. We show here that *SEP* genes may undergo functional divergence among rice, barley, and wheat. In rice, *LOFSEP* genes (*OsMADS1*, *OsMADS5* and *OsMADS34*) are predominantly involved in inflorescence branching, spikelet and floral development[48]; while *SEP3* clade genes (*OsMADS7* and *OsMADS8*) regulate inner floral organ development[49]. Unlike *OsMADS1*, which is responsible for floral organ identity specification in rice, *HvMADS1* seemed to only regulate awn length and lemma width without changing floral organ properties

**Fig. 6 | HvMADS1 physically interacts with HvAP2 and synergistically regulates downstream genes to promote awn elongation and lemma transverse growth.** **a**–**e** Interaction between MADS1 and HvAP2. Yeast two-hybrid assay (**a**): AD, GAL4 activation domain; BD, GAL4 DNA-binding domain; SD/-TL and SD/-TLHA, synthetic defined medium without Trp and Leu or without Trp, Leu, His and Ade, respectively. Split firefly luciferase complementation (SFLC) assay in tobacco leaves (**b**): Scale bar represents luminescence intensity. Bimolecular florescence complementation (BiFC) assay in tobacco epidermal cells (**c**). In vitro pull-down assay (**d**): GST-MADS1 was incubated with His-AP2, pulled down using anti-GST beads, and detected by anti-His immunoblotting. IB, Immunoblot. In vivo Co-immunoprecipitation (Co-IP) assay in tobacco leaves (**e**): GFP beads were used for immunoprecipitation (IP). **f** Effects of HvMADS1–HvAP2 interaction on HvMADS1-induced transcriptional activation (*n* = 4 biological replicates); Different letters

indicate significant differences according to one-way ANOVA with Tukey's multiple comparisons test ($p < 0.05$). Images (**g**) and statistic data of length, and cell number (**h**) of WT and *ap2* awns. Longitudinal sections of boxed regions in (**g**) are shown in Supplementary Fig. 11c. **i** Images of WT and *ap2* spikelet hulls. Red lines indicate the position of the cross-section used in (**j**, **k**). **j** Cross-sections of WT and *ap2* spikelet hulls. **k** Statistic data of the width and outer parenchyma cell number of WT and *ap2* lemmas. Statistic data of awn length (**l**), images and statistic data of lemma width (**m**) of WT, *mads1-8*, *ap2-11* and *mads1-8 ap2-11* double mutant. Values are mean ± SD, *p* values shown are from two-tailed Student's t-test; For (**a**, **b**, **c**, **d**, **e**), a representative result is shown from three independent replicates. Scale bars, 50 μm (**c**), 2 cm (**g**), 1 cm (**l**), 1 mm (**i**, **m**), and 250 μm (**j**). Source data are provided as a Source Data file.

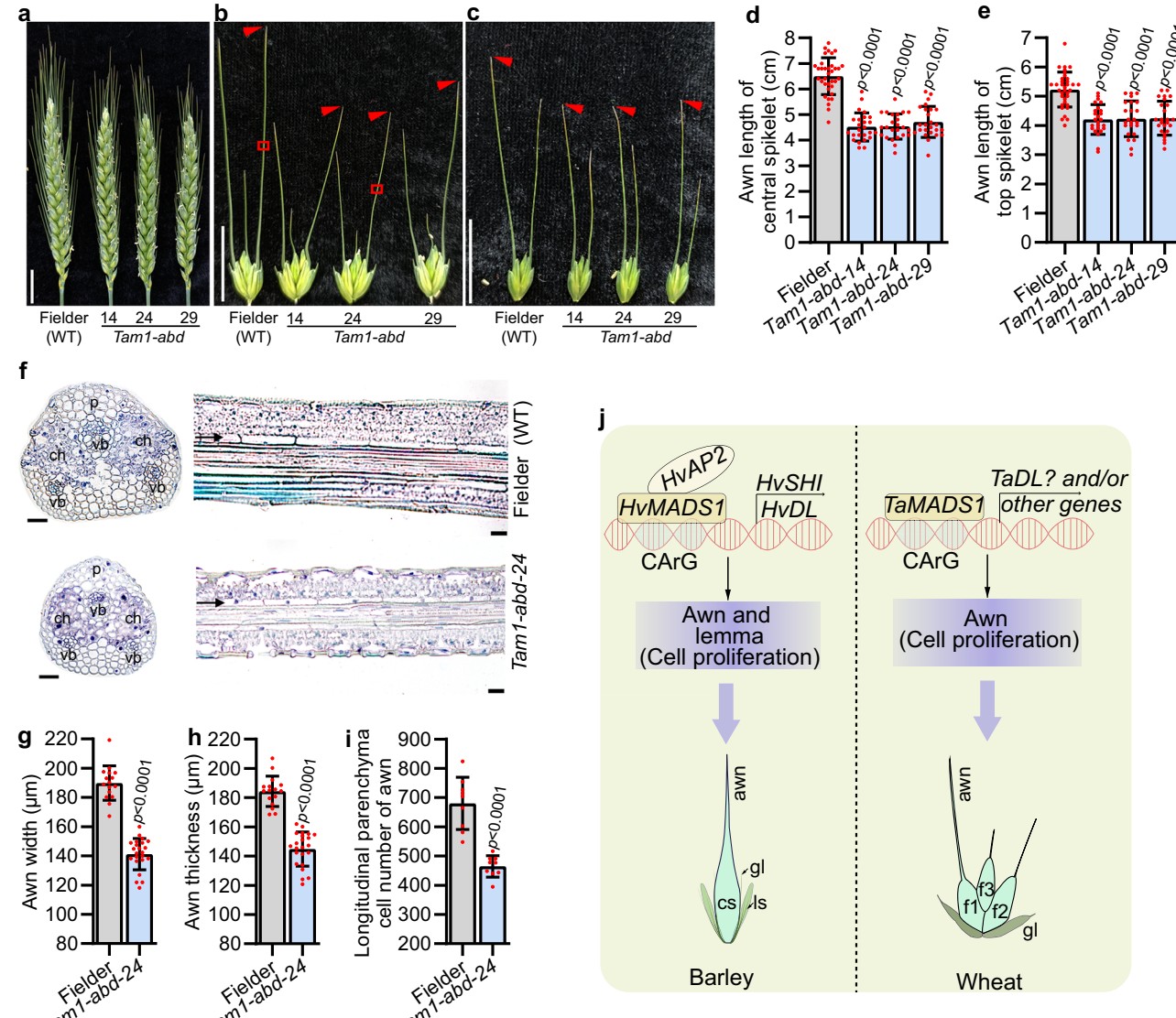

**Fig. 7 | The function of wheat *TaMADS1* in regulating awn elongation and a proposed working model for MADS1. a** Images of Fielder (WT) and three independent *Tam1-abd* mutant spikes at heading day. **b**, **c** Images of WT and *Tam1-abd* awns from the central (**b**) and top (**c**) spikelets. Statistic data of awn lengths from the central (**d**) and top (**e**) spikelets. **f** Sections from the boxed regions of (**b**) showing the size and number of cells in WT and *Tam1-abd* awns. **g**–**i** Statistic data of width (**g**), thickness (**h**), and longitudinal parenchyma cell number (**i**) of WT and *Tam1-abd* awn. **j** A proposed model for MADS1-mediated awn and lemma

development. In barley, *HvMADS1* and *HvAP2* form a complex that jointly activates potential downstream genes *HvSHI* and *HvDL*, further influencing cell proliferation and ultimately controlling the elongation of the awn and lemma transverse growth. In wheat, *TaMADS1* can also control awn elongation by influencing cell proliferation. Values are mean ± SD, *p* values obtained from two-tailed Student's t-test; Scale bars, 2 cm (**a**, **b**, **c**), and 25 μm (**f**). ch chlorenchyma tissue, f1–3 flower 1–3, cs central spikelet, gl glume, p parenchyma cells, ls lateral spikelet, vb vascular bundle. Source data are provided as a Source Data file.

in barley. This finding is consistent with our previous work that single or double mutants of barley *LOFSEP* genes (*HvMADS1*, *HvMADS5* and *HvMADS34*) have no obvious phenotypic changes in floral organs, spikelet and inflorescence branching under normal temperature[40]. There are four *SEP* genes in *Arabidopsis*, including *AtSEP3* and the *LOFSEP* clade (*AtSEP1*, *AtSEP2*, and *AtSEP4*); any single mutant shows no or weak phenotypes. However, in the quadruple *atsep1/2/3/4* mutant, all floral organs are transformed into leaf-like structures[50], and only quadruple mutant of *AtSEP4* and three MADS-box genes of other classes (*SOC1/SUPPRESSOR OF OVEREXPRESSION OF CONSTANS 1*, *SVP/SHORT VEGETATIVE PHASE* and *AGL24/AGAMOUS-LIKE 24*), namely, *atsep4/soc1/svp/agl24*, is defective in inflorescence branching[51]. The *LOFSEP* genes have undergone significant duplication in wheat; MADS1 and MADS5 are sisters to 10 and 7 wheat proteins, respectively[52]. Thus, we speculate that *SEP* genes have diverged through sub-functionalization in barley and wheat, and that some functions of barley and wheat *SEP* genes remain unidentified due to redundancy with *SEP* genes or with other classes of MADS-box genes. The *mads1/5/34* mutant, *mads1/5/34/7/8* mutant, higher order mutants of *HvSEP* genes with other barley MADS-box genes, and the equivalent mutants in wheat, will need to be created to further unravel the molecular regulatory network of *SEP* genes in *Triticeae* crops.

A combined RNA-seq and CUT&Tag approach identified multiple direct target genes of HvMADS1, including genes involved in the synthesis and signaling of hormones, such as cytokinin, gibberellins, jasmonates, and auxin, which is consistent with previous ChIP-seq data that OsMADS1 can bind to auxin-related genes[44]. In rice, auxin response factors OsETT2[30] and OsARF6[53], and the cytokinin synthesis protein OsAn-2[14,26], control awn length and/or grain size; future studies in barley could reveal whether these hormone-related genes regulate awn size and spikelet hull growth or not. Interestingly, the *mads1* mutation affected expression of multiple cell-cycle genes, and whether HvMADS1 can directly bind to promoters of these genes to regulate awn and lemma development requires further investigations.

*HvSHI*, a downstream target gene of HvMADS1, affects awn growth through regulating cell proliferation, which is in accord with the *HvMADS1* mechanism; both mutants also have a similar degree of awn shortening. Although the awn extends from the lemma tip, it is currently believed that the awn is a modified leaf, while the lemma may be modified sepal or novel organ with bract and sepal features[36,54]. Since *HvSHI* is specifically expressed in the awn primordia, it is reasonable that it affects only awn development but not lemma development, which also supports the idea that awn and lemma development are independent processes. The fact that the 1000-grain weight of *shi* grains was significantly lighter than those of WT with a 6.1% reduction (Supplementary Fig. 6d) indicate that this difference is due to the diminished photosynthetic capacity of the shorter awn, which is consistent with the results of artificial de-awning that reduces WT grain weight (Supplementary Fig. 1b). In rice, the regulatory mechanisms of grain size have been elaborated in detail[5,55], and several recent reviews have also summarized these regulatory pathways in detail[4,56]. Generally, cereal grain size is mainly determined by maternal spikelet hull, pericarp, overall pistil (including carpel), and endosperm[5,7–9,56–59]. In this study, we also demonstrated that changes in *mads1* grain weight is highly associated with changes in lemma width (Figs. 1i, 2e). We also ruled out the carpel and endosperm as the cause of the smaller *mads1* grains (Supplementary Fig. 3a–d). Thus, the 33% reduction in grain weight of *mads1* grain (Fig. 2e) is thus mainly due to the narrower lemma width; nevertheless, currently we cannot completely rule out the effect of pericarp. It is worth noting that plot sizes used in our field trials (around 2 m²/line) were much smaller than those typically used by breeders in breeding selections. In addition, our field experiments were carried out in the same region for

two consecutive years without setting up of different plot replications for the same line. Therefore, larger-scale field trials with plot replications in different regions are needed for subsequent field yield analyses.

The lemma and palea together protect the inner floral organ to ensure its stable growth and seed setting, and help to define the final grain size. The reduced seed setting rate of *mads1* plants in field trials (Supplementary Fig. 2) may be due to the inability of the narrowed lemma to enclose the palea, thus resulting in florets opening at the fertilization stage. Although ongoing research has revealed several genetic factors that control lemma development in rice[60–62], for example, *OsDL* regulates vascular patterning of the lemma[63], but its mechanism of regulating lemma development is not clearly elucidated. In barley, even less is known about factors affecting lemma development. Here, we have shown that *HvDL*, the other downstream target gene of HvMADS1, affects lemma width; the lemma width of *dl* mutant is significantly reduced (Fig. 5r), mimicking the *mads1* phenotype. Interestingly, *HvDL* also controls carpel properties (Supplementary Fig. 7g) and the number of lemma vascular bundles (Supplementary Fig. 7h), suggesting that there may be other proteins or pathways that control *HvDL* expression to regulate floral organ properties. Further investigations into upstream and downstream regulatory networks of the *HvMADS1–DL* module may help to identify the molecular regulators of floral organ development, including lemmas, in barley. Combining a series of biochemical experiments with the phenotypes of single mutants, we inferred that the binding of HvMADS1 to *HvSHI* and *HvDL* is critical for HvMADS1 in controlling awn length and lemma width. The future work should focus on the genetic analysis of double mutants (*mads1 shi* and *mads1 dl*), *mads1* mutants overexpressing *HvDL* and/or *HvSHI*, and transgenic plants with site-directed mutated motifs of *HvDL*/*HvSHI* that bound by HvMADS1, to further strengthen the understanding of the biological significance of the HvMADS1-HvSHI/HvDL module.

Previous studies have shown that *HvAP2* has pleiotropic roles in floral organ and grain development, regulating lemma identity and length as well as grain length and width[37]. In this study, we measured the width of the entire unfolded lemma rather than the curled lemma as did in other studies[37]. We found that *HvAP2* also regulates lemma transverse growth since the cell number in the transverse section of *ap2* lemma was significantly reduced, lemma width of *ap2 mads1* double mutant was narrower than those of single mutants (Fig. 6k, m). A previous study assumed that *HvAP2* could control floral development by modulating its targeting genes including *HvMADS1*[37]. Our genetic and biochemical analyses results in this study verified that *HvAP2* interacts with *HvMADS1*, forming a complex, at least in part, to participate in a common pathway promoting awn development and lemma transverse growth (Fig. 6). Interestingly, *Arabidopsis* AP2 is a repressor, which interacts with co-repressors TOPLESS to regulate downstream floral homeotic genes to control floral organ development[64]. More studies are needed to clarify the working model for HvAP2, and to answer following questions: whether HvAP2 can also act as a repressor? Is there a co-repressor similar to TOPLESS in barley? How do the two interact and thus participate in barley floral organ development?

Although grain size is a key trait for crop breeding, genes affecting grain size in an agriculturally meaningful way are still a scarce resource. In rice, a C-terminal mutation in *MADS1* forms a semidominant allele that promotes longitudinal cell proliferation in the spikelet hull, and introduction of this allele into several cultivars is favorable for both grain length and yield improvement[55,65]. Our results here suggest that *MADS1* can regulate other aspects of grain development to affect final grain size, so genome-edited superior alleles of *MADS1* will be a key resource in the future for high-yield barley breeding.

## Methods

### Plant materials and growth conditions

Barley (*Hordeum vulgare* cv. Golden Promise) was planted either in a greenhouse with a 16 h photoperiod at 18 °C/15 °C (day/night) or in a field in Minhang (31° 04′ N, 121° 45′ E; Shanghai, China) under natural conditions during November 2021–June 2022 and November 2022–June 2023. For field experiments, each WT and *mads1* mutant barley line was planted in ten rows with 0.20 m between rows, the length of each row is about 1 meter and 15 seeds were planted per row. These barley plants were grown under normal conditions and followed local agricultural practices for field management, avoiding floods, droughts, pests and diseases. 300 kg compound fertilizer per hectare (Nitrogen: Phosphorus: Potassium = 15%: 15%: 15%,) and 250 kg urea fertilizer per hectare were applied at the sowing and jointing stages, respectively. In the June of the following year, plants were randomly sampled to measure spike length, spikelet number per spike/plant, grain number per spike/plant, grain weight per plant, seed setting and thousand-grain weight. Grain weight and size were measured in grains that were treated in a 37 °C oven for one week. Wheat (*Triticum aestivum* cv. Fielder) was grown in a greenhouse with a photoperiod of 16 h at 20 °C/16 °C (day/night). For BiFC and SFLC assays, tobacco (*Nicotiana benthamiana*) was planted in a greenhouse at 23 °C with a 16 h/8 h (day/night) period.

Knockout mutants of *HvMADS1*, *HvSHI*, *HvDL*, and *HvAP2* were generated by CRISPR-Cas9 technology. For each gene, two targets were chosen, one single guide RNA (sgRNA) was driven by the *OsU6a*, and the other by the *OsU6b*; the tandem sgRNA cassette was cloned into pYLCRISPR-Cas9P$_{ubi}$-H[66]. *Agrobacterium tumefaciens* AGL1–mediated transformation of immature barley embryos[67] and selection of mutant lines were performed. A Cas9-sgRNA expression vector for *TaMADS1* knockout was constructed and introduced into *Agrobacterium* EHA105 for transformation of immature wheat embryos[68]. Homozygous progeny plants (≥T$_2$ generation) were used in the analyses. Primers used for constructs are listed in Supplementary Data 4.

### Morphological and histological analysis

Barley grains and spikelets were collected from the central parts of the main spike of at least ten individual plants per genotype for phenotyping analysis. For wheat, at least twenty-four spikelets from main spike were collected from the central and top parts of at least eight individual plants for each genotype for phenotypic observation. Barley spike photosynthetic gas exchange parameters were measured as described[69] with minor modifications. Briefly, the entire awn on all spikelets of uniform main spikes of healthy WT plants at heading were cut off using sharp blades, and the whole-spike net photosynthetic rate was measured three days after de-awning to rule out the effect of wounding. The grain weight of both WT and de-awned WT was measured at harvest time. For SEM analysis, fresh awn, spikelet, and inflorescence were fixed in fresh FAA (3.5% formalin, 5% acetic acid and 50% ethanol) for 24 h. After being dehydrated in a graded ethanol series, samples were dried and coated with gold using Leica EM SCD050 sputtering device, then photographed with a scanning electron microscope (Hitachi, S-3400N). For paraffin sectioning, after being dehydrated and infiltrated by Histo-Clear II and ethanol, samples were transferred into paraffin, sectioned at 8 μm thickness, stained with toluidine blue, and observed under the light microscope (Nikon, ECLIPSE 80i). ImageJ software was used to calculate cell length, cell width and cell number, and at least 9 individual samples were used for such analyses. For the length and number of awn cells, the awn was longitudinally cut at the middle part of the awn, cell length per awn across the cut entire awn region was measured, and the resulting value was divided by the awn length to yield cell numbers per awn. For the size and number of lemma cells, lemma was cut transversely at the middle part of the lemma, and cell length and cell numbers were directly measured and counted, respectively.

### In situ mRNA hybridization

Fresh developing inflorescences were cut and immediately fixed in FAA solution for 6 h, then dehydrated, infiltrated, embedded, and sectioned as described above for paraffin sectioning. RNA probes (sense and anti-sense) were labelled using the SP6/T7 DIG RNA labeling kit (Roche) according to manufacturer's instructions, and listed in Supplementary Data 4. Pre-treatment of sections, hybridization, and immunological detection were performed[70].

### RT-qPCR and RNA-seq analysis

Lemma and awn from spikelets in the central part of stage W7.5 spike from WT and *mads1* mutants were collected for RNA-seq and RT-qPCR. In addition, whole shoot (7 days after germination), developing inflorescences at W2.0–W9.5, mature lemma, awn, palea, and developing grains from 0–35 days after fertilization (DAF) were collected from WT and *mads1* plants for spatiotemporal expression analysis of *HvMADS1*, *HvSHI*, *HvDL* and *HvAP2* expression. All tissues collected contain three biological repeats, each is a pooled tissue from 15-30 individual plants. Total RNA was isolated using TRIzol Universal Reagent (Tiangen) and cDNA synthesized using the Tiangen FastQuant RT Kit according to manufacturer's instructions. Quantitative real-time PCR (RT-qPCR) assays were performed using QuantiNova SYBR Green PCR Kit (Qiagen). Expression of *HvActin7* was used as the internal control[71] and primers used for RT-qPCR are all listed in Supplementary Data 4.

For transcriptome analysis (RNA-seq), library construction and sequencing were performed by BGI (Wuhan, China) using an Illumina 2500 platform. After screening and trimming, clean reads were mapped to the barley reference genome, and transcript levels calculated as fragments per kilobase million (FPKM) using RSEM (v1.3.3). Differentially expressed genes (DEGs) were selected using R package DESeq2 (v.1.30.0) with a criterion of |fold change| ≥ 1.5 and false discovery rate <0.05. DEGs were analyzed for enriched pathways by Gene Ontology (GO) using AgriGO2[72]. REVIGO was used to summarize and visualize the GO term results[73].

### Cut&Tag and ChIP-qPCR

Cut&Tag treatments were performed as described[74] with minor modifications. Nuclei were extracted from W7.5 lemma and awn of *pro::HvMADS1−eGFP* transgenic plants (created in our previous work[40]), each with two biological replicates. Primary antibodies were anti-GFP (ab290, Abcam) and IgG (as control; 12-370, Sigma−Aldrich), and the guinea pig anti-Rabbit IgG (PAB9407, Abnova) was used as secondary antibody. The tagmentation assay was performed after use of Tn5 transposase according to the manufacturer's instructions (Vazyme, S603-01). Libraries were prepared using the TruePrep Index Kit V2 for Illumina (Vazyme, TD202) and NEBNext Hi-Fi mix (NEB, M0541S), purified with AMPure beads (Beckman, A63881), and sequenced at Personal Biotechnology (Shanghai) with an Illumina Nova. Cut&Tag data processing and analysis were performed based on the existing pipeline[74,75].

The ChIP-qPCR assay was performed as reported with minor modifications[76]. Approximately 2 g of W7.5 inflorescences from WT and *pro::HvMADS1−eGFP* plants were cross-linked with 1% (v/v) formaldehyde and ground in liquid nitrogen. The chromatin was sonicated, then the DNA fragments were incubated with GFP-Trap Magnetic Agarose (ChromoTek, Munich). Immunoprecipitated DNA was analyzed by quantitative PCR using *HvSHI* and *HvDL* primers (Supplementary Data 4). Samples prior to precipitation were used as input controls, and the final result was a comparison of the enrichment levels of target genes between WT and GFP lines.

## Electrophoretic mobility shift assay (EMSA)

The complete *MADS1* coding sequence (CDS) was fused to 6×His and cloned into the pGADT7 vector for in vitro protein synthesis using the TNT T7/SP6 Wheat Germ Protein Expression System (Promega). Briefly, the recombinant MADS1−6×His protein was incubated with fluorescein amidite (FAM)-labeled probe and competition probe in binding buffer for 30 min at 25 °C. Samples were loaded onto a 6% native polyacrylamide gel at 4 °C, and the FAM signals were detected by the Cy2 channel of a ChemiDoc MP imaging system (BioRad). All probe sequences are listed in Supplementary Data 4.

## Dual-luciferase (dual-LUC) and transcriptional activity assays in barley protoplasts

The preparation and transformation of barley protoplasts[77] were performed. For dual-LUC assays, the *HvSHI* and *HvDL* promoters were amplified and cloned into pGreenII-0800-LUC to drive luciferase (*LUC*) expression as reporters. Effectors were created by ligating *HvMADS1* and *HvAP2* CDS into the pGreenII-0000 construct. Reporter and effector vectors were co-transformed into barley protoplasts via polyethylene glycol-mediated transformation[78], and cultured overnight at 22 °C in the dark. LUC activity was measured using a Dual-Luciferase Reporter Assay System (Promega) with Renilla luciferase (REN) as an internal control.

For the transcriptional activity assay, *5×GAL4-LUC* and *3SS::REN* were used as the reporter and the internal control, respectively. *3SS::HvAP2* or *3SS::GFP* were used as plus vectors. Full-length CDS of *MADS1* and *VP16* (constitutive transcriptional activator) were fused to *GAL4BD* to generate effector vectors. Protoplast transformation and LUC/REN measurement followed the same protocols as for dual-LUC assays.

## Yeast two-hybrid assays

Yeast two-hybrid assays were performed following manufacturer's instructions (Clontech). The full-length or truncated CDS of *HvMADS1* and of *HvAP2* were amplified and cloned into inserted into pGADT7 and pGBKT7, respectively. Different combinations of these vectors were co-transformed into yeast strain AH109 and selected on SD-Leu/-Trp or SD-His/-Ade/-Leu/-Trp for 2–3 days.

## Split firefly luciferase complementation (SFLC) and bimolecular fluorescence complementation (BiFC) assays

For SFLC assays, *HvMADS1* and *HvAP2* CDS was cloned into the pCAMBIA1300-nLUC and pCAMBIA1300-cLUC vectors[79], respectively. For BiFC assays, *HvMADS1* and *HvAP2* CDS was ligated into pXY104-cYFP and pXY106-nYFP[80], respectively. Recombinant constructs were transformed into *A. tumefaciens* GV3101 and different combinations of *Agrobacterium* were co-infiltrated into tobacco leaves. Subsequently, the plants were grown in the dark for 48 h, and the activated luciferase reconstitution signals were captured with Tanon 5200 imaging system after injecting 1 mM luciferin to leaf regions. The YFP fluorescence signal was observed under a Leica SP5 confocal laser-scanning microscope.

## In vitro pull-down assay

The pull-down assay was performed as described in the *Glutathione Affinity Handbook* (Qiagen, 3rd Edition). The *HvMADS1* and *HvAP2* CDS was cloned into pGEX4T1 and pET-32a to produce *GST-MADS1* and *His-AP2*, respectively. The GST-MADS1 and His-AP2 recombinant proteins were expressed in *Escherichia coli* BL21 (*DE3*), and purified with Glutathione HiCap Matrix resin (Qiagen) and Ni-NTA Agarose (Qiagen), respectively. GST-MADS1 and GST were incubated with glutathione beads, respectively, and then His-AP2 or His added to the reaction, followed by incubation overnight at 4 °C. Beads were washed and the eluted proteins were separated by 10% SDS-PAGE, and finally immunoblotted and probed with anti-GST and anti-His antibody, respectively.

## In vivo co-immunoprecipitation (Co-IP) assay

Co-IP assays were performed as described[81] with minor modifications The *35S::MADS1-HA* and *35S::YFP-AP2* plasmids in *Agrobacterium GV3101* were transiently co-expressed into tobacco leaves as described above. The empty-vector *35S::YFP* was used as a negative control. Proteins were isolated with extraction buffer containing 10 mM Tris−HCl at pH 7.5, 150 mM NaCl, 0.5 mM EDTA, 0.2 % TrixtonX-100, 0.2 % NP-40, 1 mM DTT, 2% (v/v) PVP40, 0.1% (v/v) Tween-20, and 1×protease inhibitor cocktail (Roche). A small portion of the protein extract was saved as input and the rest incubated with GFP beads (GFP-Trap® Agarose, Chromo Tek) for 4 h at 4 °C. Beads were then denatured in protein loading buffer, and subjected to SDS-PAGE. Anti-GFP (G1544, Sigma) and anti-HA (M20003, Abmart) antibodies were used to detect protein interaction signals.

## Phylogenetic analyses

Homologous proteins in other species, including wheat, *Brachypodium*, sorghum, and rice, were obtained using the HvMADS1 (HOR-VU4Hr1G067680) protein sequence to query against the EnsemblPlants database (http://plants.ensembl.org). The amino acid alignment of these MADS-box proteins was imported into MEGA5 using the neighbor-joining method to generate a phylogenetic tree with bootstrap values of 1000 replicates.

## Statistics & Reproducibility

Statistical analyses of all bar graphs were performed using GraphPad Prism (Version 8.0.2) (https://www.graphpad.com/scientific-software/prism/) or Microsoft Excel (2016). Values were represented as mean ± SD, *p* values are made by two-tailed Student's t-test or one-way ANOVA with Tukey's multiple comparisons test. The biological replicates of experiments presented in this study are indicated in the respective figure legends. No data were excluded from the analyses and no statistical method was used to predetermine sample size. The experiments were not randomized and the investigators were not blinded to allocation during experiments and outcome assessment.

## Reporting summary

Further information on research design is available in the Nature Portfolio Reporting Summary linked to this article.

# Data availability

The transcriptomic data and CUT&Tag data generated in this study have been deposited in the National Center for Biotechnology Information under accession code GSE228410 and PRJNA1012547, respectively. All data supporting the conclusions of this study are presented within the paper and its Supplementary Information files. Source data are provided with this paper.

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

## Acknowledgements

The authors would like to dedicate this paper to Prof. Dabing Zhang whose background work and ideas were the basis of this project and who tragically passed away in June 2023. The authors would also like to thank Dr. Xuelei Lin and Prof. Jun Xiao (Institute of Genetics and Developmental Biology, Chinese Academy of Sciences) for their technical assistance with the CUT&Tag assays. We thank Dr. Natalie Betts for editing this manuscript, Prof. Genying Li (Crop Research Institute, Shandong Academy of Agricultural Sciences) for creating the wheat *mads1* mutant lines, Prof. Chengdao Li (Murdoch University) for performing the analysis of natural variation in *HvMADS1*, Prof. Yaoguang Liu (South China Agricultural University) for providing CRISPR-Cas9 vectors, and Mr. Ting Luo (Shanghai Jiao Tong University) for helping with barley cultivation. This research was funded by grants from the Australian Research Council (DP210100956 and DP230102476 to D.Z.); the Australian Research Council Training Centre for Accelerated Future Crop Development (IC210100047 to D.Z.); the Australian Research Council Linkage (LP210301062 to D.Z.); the National Natural Science Foundation of China (32130006 and 31970803 to D.Z.); the Yazhou Bay Seed Laboratory Project (B21HJ8104 to D.Z.); the Innovative Research Team (Ministry of Education); and the 111 Project (B14016 to D.Z.).

## Author contributions

D.Z. conceived the project and supervised the study. D.Z., Y.Z., G.L. and J.X.S designed experiments. Y.Z. and C.S. performed most of the experiments and analyzed data. J.S. performed bioinformatics analyses. Y.Y. helped to evaluate some phenotypic data. L.Y. created the barley *shi* and *ap2* mutant lines. Q.S. conducted spike photosynthesis analysis. C.S. and Y.Z. performed CUT&Tag assays. Y.Z., C.S., and D.Z. wrote the manuscript with comments from all authors. J.X.S revised the manuscript. All authors read and approved the manuscript.

## Competing interests

The authors declare no competing interests.
