## [Peer Review File · Nature Communications]

MADS1-regulated lemma and awn development benefits
barley yieldREVIEWER COMMENTS

Reviewer #1 (Remarks to the Author):

In the manuscript by Zhang, et al., the authors determine the function of the E-class gene, HvMADS1 from barley with respect to lemma and awn development. While the ABCDE model of flower and floral organ development has been well described in model plants such as Arabidopsis and snapdragon, less is known about ABCDE genes and the transcription factors they encode, from other species, including barley and wheat. The authors show that HvMADS1 positively regulates lemma width and awn length and promotes cell proliferation. Using CUT&Tag and RNA-seq, the authors identify likely direct targets of HvMADS1, including HvSHI and HvDL. CRISPR ko of these genes resulted in similar phenotypes to the lof HvMADS1. The authors identify a protein interacting partner of HvMADS1, HvAP2, an A class gene. This work provides important characterization of a barley MADS TF with respect to organ development and that plays a critical role in grain size and yield, thus of interest for crop improvement. The authors perform an impressive number of different experiments and provide new information as to the function of an important MADS class E gene in barley.

Major points

What proteins were tested in the Y2H assays? All the ABCDE encoded proteins? Were other positives found as AP2 is not a MADS TF but the other A,B,C D class are. Since the E class SEPALLATAs from Arabidopsis, for example, do not interact directly with AP2, that I recall, this is an unusual finding and should be discussed. Were other A, B, C and E class TFs shown to interact as well? This would be expected. It is not clear what was tested and if only HvAP2 was tested what was the rationale for this?

AP2 is a repressor, at least in Arabidopsis and interacts with TOPLESS co-repressor, for example to repress the C class gene AGAMOUS. Is there a similar function for HvAP2?

Can the authors perform an EMSA with HvMADS1 and HvAP2 using the HvSHI and/or HvDL promoter DNAs? Since these are supposed to form a transcriptionally active complex, there should be a supershift. This would further support the interaction and DNA binding of HvMADS1 and HvAP2.

Minor points discussion-

I do not understand what “three MADS-box genes of other classes” refers to. What other classes?

335 “However, the quadruple atsep1/2/3/4 mutant had a strong phenotype, with all floral organs transformed into leaf-like structures, and only quadruple mutants of AtSEP4 and three MADS-box genes of other classes had defects in inflorescence branching. “

Reviewer #2 (Remarks to the Author):

Zhang et al's (2023) well-crafted manuscript describes the function of the barley transcription factor-encoding gene HvMADS1. This paper builds on their previous work showing that loss of HvMADS1 function causes changes in reproductive architecture (Li et al., 2021). Here they focus on HvMADS1' control of awn and lemma growth. They demonstrate that hv mads1

mutants have shorter awns and narrower lemmas, which in both cases may result from reduced cell number, suggesting that HvMADS1 promotes cell proliferation during organ development. They then associate these differences with reductions in spike photosynthetic rate as well as reduced grain width, depth and weight in the mutant. In a significant advance, they use comparative transcriptomics and CUT&Tag assays to reveal multiple direct genomic targets of HvMADS1, including HvDROOPING LEAF (HvDL) and HvSHORTENED INTERNODES, genes previously shown to promote awn development in rice and barley, respectively. They test the functional relevance of these targets by generating and characterising loss of function *hvd1* and *hvshi* alleles which phenocopied multiple aspects of the *hvmads1* phenotype, suggesting that HvMADS1 works through these targets to control lemma and awn development. In addition, they show that HvMADS1 interacts with HvAPETALA2, a factor which regulates similar traits, and that HvMADS1's transcriptional activation may increase in the presence of HvAP2. Lastly, gene editing MADS1 in wheat suggests that MADS1's role in controlling awn length are conserved between these species. Taken together, they reveal that how HvMADS1 may work as an upstream regulator of a pathway controlling awn and lemma development which is correlated with final grain parameters and yield in barley.

This paper's combination of molecular and functional insight about the genes important for lemma and awn growth provides a major advance in our understanding of development of these grass-specific floral organs in temperate cereals, especially with regards to the targeting of HvDL and HvSHI by HvMADS1, as the upstream regulation of these genes is not well understood in grass models.

However, some results in this paper overlap with existing data and some ideas published in Shoesmith et al (2021), including the HvMADS1 in situ as well as the floral and grain phenotypes due to a loss of HvAP2 function, which could have been better cited with reference to their own results. Furthermore, this paper repeats some information from their previous manuscript Li et al (2021) which showed both the short awns and "retarded awn elongation in the mutant (Extended Data Fig. 5a,b)", although the SEM images in this manuscript are different.

Overall, most data appear robust, convincing, and thorough. The CUT&Tag assays and the gene-edited *hvd1* mutants provided particularly strong supporting data for their conclusions. However, I do have a couple major concerns/suggestions:

- i) One major concern is the manuscript's message that grain differences due to a loss of HvMADS1 function are caused by changes in lemma/ awn growth. At multiple points in the manuscript, grain changes are described as being caused by the changes in lemma development (eg. line 26 "HvMADS1 positively regulates awn length and lemma width, leading to increased grain size and weight"). However, they only show a correlation and do not provide evidence for a causative relationship. This is also a concern with the research in rice cited in their paper. Alternative or additional possibilities should be addressed or considered, such as a role for HvMADS1 in the carpel, caryopsis or grain which could contribute to the grain differences, especially as HvMADS1 is relatively highly expressed in developing carpels and caryopses. Furthermore, the putative direct target of MADS1, HvDL clearly plays a major role in the carpel according to their manuscript (line 250-251 "*hvd1* mutants were unable to set seeds due to defective carpels"), consistent with its well-known role in rice to specify the carpel (Yamaguchi et al., 2004). Thus, both HvMADS1 and HvDL could influence carpel characteristics.
- ii) They state that HvMADS1 controls awn and lemma development by directly modulating HvSHI and HvDL. Their gene-edited lines strongly support that HvDL promotes awn elongation and lemma lateral growth, and it was already known from loss of HvSHI function *lks1* alleles (Yuo et al, 2012) that HvSHI promotes awn elongation, and this paper provides

robust support that HvMADS1 directly binds to these target genes. These together are suggestive that the binding relationship is important for HvMADS1 control of awn length and lemma width. However, without genetic analyses to demonstrate how this relationship impacts the phenotypes of double mutants (eg. wrt epistasis), or transgenic rescue by overexpression of HvDL or HvSHI in the *hvmads1* mutants, we do not have an estimate of the importance of this binding (or loss thereof) to the phenotypes observed in *hvmads1*. So, I would tone this conclusion down.

I also have a few more minor concerns/suggestions:

- i) I couldn't find detailed information about the exact gene edits for the HvMADS1, HvDL, HvSHI and HvAP2 alleles described in the paper, including the changes within each allele, the screening, generation analysed or Cas9 presence/absence. It is also unclear if the *hvmads1* alleles used in this paper are the same ones described in their previous publication (Li et al., 2021) – potentially *hvmads1-5* is?
- ii) More explanation of the awn removal experiment is needed. When/ how were the spikes de-awned and how long after were the photosynthetic rates determined? Also, were grain from these spikes compared with awned spikes in either genotype?
- iii) Their conclusion that HvMADS1 promotes growth through cell proliferation is primarily based on 'cell number' estimates extrapolated from the measurements of cell size within a small section of the awn and lemma. They do not discuss the possibility that cell size might change throughout the organ which could also contribute to the awn length and lemma width differences – so I would use caution the interpretation to extrapolate across the entire organ and more directly state that the cell number differences were extrapolated from the measurements of cell size within a small section of the awn and lemma.
- iv) The last sentence in their discussion suggests that 'other superior alleles of HvMADS1, natural or gene-edited, will be a key resource in the future for high-yield barley breeding'. Their previous study (Li et al., 2021) showed high levels of conservation in HvMADS1 across barley, so it is unclear how likely natural superior alleles would be found.
- v) Introduction incorrectly states that SHORT INTERNODES is the only gene reported in barley to control awn elongation. Shoosmith et al (2021) demonstrated that loss of APETALA2 function leads to shortened lemma awns.
- vi) I'm not convinced that the H4 in situ shows a gradual attenuation of signal, although the awn itself is obviously smaller.
- vii) Could the increase in Shi::LUC expression when AP2 and MADS1 are co-expressed be explained by an additive effect? i.e. both AP2 and MADS1 binding separate reporter constructs – unless you think the proteins are provided in excess to the availability of the reporter? Something to consider.

Below are a couple issues with data presentation:

- i) Figure 3b far right-hand graph in panel describing the outer parenchyma cell number – was this determined from the SEM or a tissue section. This wasn't clear.
- ii) Figure 5a and 5j – I'm not clear what the RNAseq panel is showing – does this plot represent reads from either genotype indicating a loss of reads in the mutant? Also, both panels are too small to read easily.
- iii) Figure 5n 'M' from 'MADS1' is missing as an x-axis label. Furthermore, the figure caption does not describe co-expressed AP2 being part of this experiment.

Reviewer #3 (Remarks to the Author):

Zhang et al. study the effect of knockout mutations of barley E-class homeobox gene

HvMADS1 on awn, lemma and seed development, inspired by the effects of similar analyses in rice. They could demonstrate that HvMADS1 is involved in awn/lemma development, likely through the interaction with HvSHI and HvAP2. The work builds on previous studies and takes advantage of a number of molecular biology tools, e.g. MADS1-GFP fusions which were required for the cut&tag analysis. The effect on awn length was reproduced for triple KO mutants of the putative orthologous genes in wheat.

Overall the study looks convincing and due to the similarities but also contrasts between rice, barley and wheat it is likely that the study will be interesting to a broader molecular plant biology/developmental biology audience.

My main concerns are about the yield claims. I think authors should be very careful with making strong yield claims without documenting proper conclusive field experiments. There is no information on the size and the design of the field experiments hence the statistical analysis provided for yield parameters is non-conclusive.

Methods are too brief for other sections, e.g. no information is available on sample size for all morphological and cell biology experiments. In the molecular experiments the definition of replicates needs to be explained and defined. To my opinion the authors have worked exclusively with technical controls.

data availability: the authors say that "Source data are provided with this paper. Additional data supporting the findings of this study are available from the corresponding author upon reasonable request". This is basically not true and the manuscript must not be accepted without free access to all sequencing raw data which must be submitted to public repositories. Plant material should be provided with a proper Biosample ID. The notion "available upon reasonable request" should never be accepted because every published result must be open to reproduction, therefore materials must be made available without any restriction.

Point-by-point responses to the comments by Reviewers

Response to comments of Reviewer #1

Overall comment: In the manuscript by Zhang, et al., the authors determine the function of the E-class gene, HvMADS1 from barley with respect to lemma and awn development. While the ABCDE model of flower and floral organ development has been well described in model plants such as Arabidopsis and snapdragon, less is known about ABCDE genes and the transcription factors they encode, from other species, including barley and wheat. The authors show that HvMADS1 positively regulates lemma width and awn length and promotes cell proliferation. Using CUT&Tag and RNA-seq, the authors identify likely direct targets of HvMADS1, including HvSHI and HvDL. CRISPR ko of these genes resulted in similar phenotypes to the lof HvMADS1. The authors identify a protein interacting partner of HvMADS1, HvAP2, an A class gene. This work provides important characterization of a barley MADS TF with respect to organ development and that plays a critical role in grain size and yield, thus of interest for crop improvement. The authors perform an impressive number of different experiments and provide new information as to the function of an important MADS class E gene in barley.

Answer: We appreciated your supportive and positive comments on our work.

Comment 1: What proteins were tested in the Y2H assays? All the ABCDE encoded proteins? Were other positives found as AP2 is not a MADS TF but the other A,B,C D class are. Since the E class SEPALLATAs from Arabidopsis, for example, do not interact directly with AP2, that I recall, this is an unusual finding and should be discussed. Were other A, B, C and E class TFs shown to interact as well? This would be expected. It is not clear what was tested and if only HvAP2 was tested what was the rationale for this?

Answer: Many thanks for raising such a critical comment and sorry for not providing relevant information in our previous version. Indeed, we tested interactions of HvMADS1 with all ABCDE-class proteins, including AP2, in the Y2H assays, and

found that HvMADS1 can interact with other A-, B-, C-, D-, E-class, and AGL6 subfamily homeotic proteins (except HvMADS4 and HvMADS14), which is consistent with the results reported in *Arabidopsis* (Theißen et al., 2016) and rice (Hu et al., 2015). Notably, we also found that HvMADS1 interacts with HvAP2, a known regulator of awn development in barley (Shoesmith et al., 2015) to promote awn development and lemma transverse growth (lines 284-296; Fig. 6), which has not been reported in *Arabidopsis* and rice. Based on these, we selected AP2 for further biochemical and genetic validations. We have revised this part in Results section (lines 281-286) and presented Y-2-H results in Supplementary Fig. 9.

Comment 2: AP2 is a repressor, at least in *Arabidopsis* and interacts with TOPLESS co-repressor, for example to repress the C class gene AGAMOUS. Is there a similar function for HvAP2?

Answer: Thanks for this interesting comment.

Our Y2H assay did find that HvAP2 also interacts with HvTOPLESS (Figure-1 below) as its orthologs AP2 in *Arabidopsis* does (Krogan et al., 2012).

Figure-1. **Interaction of HvAP2 with HvTPL.** A representative result is shown from three independent replicates. AD, pGADT7; BD, pGBKT7; SD-LT, selective media without Trp and Leu; SD-LTHA, selective media without Trp, Leu, His and Ade.

In addition, Our RT-qPCR analysis showed that the expression of *HvMADS3* but not that of *HvMADS58* is significantly increased in lemma and awn of *ap2* mutant during W7.5 and W8.5 stage relative to the WT (Figure-2).

Figure-2. Expression of *HvMADS3* and *HvMADS58* in lemma and awn of *ap2* mutants

Currently, we do not know the genetic and molecular relationships between *HvAP2* and *HvMADS3* in barley, neither are the involvement of *HvTOPLESS* and the biological functions of the interaction and regulation. To answer these questions, more explorations are needed to clarify the working model for *HvAP2*, which is beyond the scope of this manuscript, which will be done in another project.

As suggested, we have discussed this issue in the Discussion section (lines 445-453).

Comment 3: Can the authors perform an EMSA with *HvMADS1* and *HvAP2* using the *HvSHI* and/or *HvDL* promoter DNAs? Since these are supposed to form a transcriptionally active complex, there should be a supershift. This would further support the interaction and DNA binding of *HvMADS1* and *HvAP2*.

Answer: We appreciated very much for this comment.

We do agree with your comments that the binding of the transcriptionally active complex formed by *HvMADS1* and *HvAP2* to downstream DNA was only confirmed by transcriptional activation and transient dual luciferase assays in barley protoplasts in our manuscript. However, we have, indeed, tried many times to simultaneously add *HvAP2* and *HvMADS1* into the EMSA system, we also tried different protein purification methods and different binding buffers, but none of them showed supershift. We hypothesize that the binding of the complex of *HvAP2* and *HvMADS1* to

downstream DNA may still require unknown specific conditions.

In the revised version, we have added data of *ap2 mads1* double mutant in Fig. 6 l and m and in the Result section (lines 307-314) to further clarify their genetic interactions.

Comment 4: Minor points discussion I do not understand what “three MADS-box genes of other classes” refers to. What other classes?

335 “However, the quadruple *atsep1/2/3/4* mutant had a strong phenotype, with all floral organs transformed into leaf-like structures, and only quadruple mutants of *AtSEP4* and three MADS-box genes of other classes had defects in inflorescence branching. “

Answer: Thank you so much for the careful review.

We are so sorry that we did not clarify it clearly in the previous version. We have revised this part as follows: “However, in the quadruple *atsep1/2/3/4* mutant, all floral organs are transformed into leaf-like structures, and only quadruple mutant of *AtSEP4* and three MADS-box genes of other classes (*SOC1/SUPPRESSOR OF OVEREXPRESSION OF CONSTANS 1*, *SVP/SHORT VEGETATIVE PHASE* and *AGL24/AGAMOUS-LIKE 24*), namely, *atsep4/soc1/svp/agl24*, is defective in inflorescence branching” (lines 367-371).

Response to comments of Reviewer #2:

Overall comment: Zhang et al’s (2023) well-crafted manuscript describes the function of the barley transcription factor-encoding gene *HvMADS1*. This paper builds on their previous work showing that loss of *HvMADS1* function causes changes in reproductive architecture (Li et al., 2021). Here they focus on *HvMADS1*’ control of awn and lemma growth. They demonstrate that *hvmads1* mutants have shorter awns and narrower lemmas, which in both cases may result from reduced cell number, suggesting that *HvMADS1* promotes cell proliferation during organ development. They then associate

these differences with reductions in spike photosynthetic rate as well as reduced grain width, depth and weight in the mutant. In a significant advance, they use comparative transcriptomics and CUT&Tag assays to reveal multiple direct genomic targets of HvMADS1, including HvDROOPING LEAF (HvDL) and HvSHORTENED INTERNODES, genes previously shown to promote awn development in rice and barley, respectively. They test the functional relevance of these targets by generating and characterising loss of function *hvd1* and *hvs1* alleles which phenocopied multiple aspects of the *hvmads1* phenotype, suggesting that HvMADS1 works through these targets to control lemma and awn development. In addition, they show that HvMADS1 interacts with HvAPETALA2, a factor which regulates similar traits, and that HvMADS1's transcriptional activation may increase in the presence of HvAP2. Lastly, gene editing MADS1 in wheat suggests that MADS1's role in controlling awn length are conserved between these species. Taken together, they reveal that how HvMADS1 may work as an upstream regulator of a pathway controlling awn and lemma development which is correlated with final grain parameters and yield in barley.

This paper's combination of molecular and functional insight about the genes important for lemma and awn growth provides a major advance in our understanding of development of these grass-specific floral organs in temperate cereals, especially with regards to the targeting of HvDL and HvSHI by HvMADS1, as the upstream regulation of these genes is not well understood in grass models.

Answer: We deeply appreciated your positive comments and great support on our manuscript.

Comment 1: However, some results in this paper overlap with existing data and some ideas published in Shoesmith et al (2021), including the HvMADS1 in situ as well as the floral and grain phenotypes due to a loss of HvAP2 function, which could have been better cited with reference to their own results.

Answer: We appreciated very much for this comment.

We have revised and cited relevant references as suggested. For *HvAP2*, we have cited

relevant article in the Introduction section “In barley, *SHORT INTERNODES (SHI)* controls awn elongation and pistil morphology and *APETALA2 (AP2)* regulates awn development and awn-lemma boundary” (lines 85-87) and in the Discussion section “Previous studies have shown that *HvAP2* has pleiotropic roles in floral organ and grain development, regulating lemma longitudinal and pericarp transversal development and thus grain length and width” (lines 439-441). In addition, we have revised the manuscript in the Results section regarding awn and grain size of *ap2* mutants as “The observed phenotypes of awn length and grain size of *ap2* mutants are consistent with previous reports” (lines 305-307).

For *HvMADS1* short awn phenotype, we described short awn phenotype in our mutants and cited previously reported as “All three homozygous mutant plants also exhibited significantly shorter awns than WT plants (Fig. 1a, d), confirming an important role of *HvMADS1* in awn development as previously reported” (lines 125-128). For *HvMADS1 in situ* hybridization, Shoemith’s work showed that the expression of *MADS1* in young and old spikelets particularly in developing glumes and lemma/palea of the Bowman background is relative weak; here, we provided a clear and strong *in situ* expression pattern of *MADS1* in the developing inflorescence of Golden Promise background (Fig. 3c).

Comment 2: Furthermore, this paper repeats some information from their previous manuscript Li et al (2021) which showed both the short awns and “retarded awn elongation in the mutant (Extended Data Fig. 5a,b)”, although the SEM images in this manuscript are different.

Answer: We appreciated very much for this suggestive comment to improve our manuscript quality.

We have revised this part thoroughly and clarified that the *HvMADS1* alleles used in this study is different from previously reported one (Li et al., 2021) in the Results section as “We generated three knockout lines of *HvMADS1* using CRISPR-Cas9

system, namely *mads1-3*, *mads1-5* and *mads1-8* in barley cv. Golden Promise (Supplementary Fig. 1a), with different alleles from those reported in previous studies. All three homozygous mutant plants also exhibited significantly shorter awns than WT plants (Fig. 1a, d), confirming an important role of *HvMADS1* in awn development as previously reported” (lines 123-128).

Therefore, although our work was an extension of Li’s to reveal underlying molecular mechanisms, the *HvMADS1* alleles used in our study is different from that used by Li. In addition, Li’s work was carried out in an Australian laboratory, ours was done independently in a Chinese laboratory.

Comment 3: One major concern is the manuscript’s message that grain differences due to a loss of *HvMADS1* function are caused by changes in lemma/ awn growth. At multiple points in the manuscript, grain changes are described as being caused by the changes in lemma development (eg. line 26 “*HvMADS1* positively regulates awn length and lemma width, leading to increased grain size and weight”). However, they only show a correlation and do not provide evidence for a causative relationship, This is also a concern with the research in rice cited in their paper. Alternative or additional possibilities should be addressed or considered, such as a role for *HvMADS1* in the carpel, caryopsis or grain which could contribute to the grain differences, especially as *HvMADS1* is relatively highly expressed in developing carpels and caryopses. Furthermore, the putative direct target of *MADS1*, *HvDL* clearly plays a major role in the carpel according to their manuscript (line 250-251 “*hvd1* mutants were unable to set seeds due to defective carpels”), consistent with its well-known role in rice to specify the carpel (Yamaguchi et al., 2004). Thus, both *HvMADS1* and *HvDL* could influence carpel characteristics.

Answer: This is a wonderful comment that helps us a lot to clarify the function of *HvMADS1* in grain size.

To address your concerns, we have performed additional analyses, provided more

results, and revised the corresponding parts.

For carpel: 1) we have compared the carpel size between WT and *mads1* mutant, added the results in Supplementary Fig. 3, and revised the Results part as “To rule out the effect of carpel size on grain size, we also compared carpels size between WT and *mads1* mutant, and found no significant difference in the carpels size between WT and mutant at W10.0 stage (Supplementary Fig. 3a, b)” (lines 168-171); 2) We also added the carpel phenotype of *dl* mutant in Supplementary Fig. 7h, and revised corresponding part in the Discussion part as “Interestingly, *HvDL* also controls carpel properties (Supplementary Fig. 7h) and the number of lemma vascular bundles (Supplementary Fig. 10i), suggesting that there may be other proteins or pathways that control *HvDL* expression to regulate floral organ properties. Further investigations into upstream and downstream regulatory networks of the HvMADS1–DL module may help to identify the molecular regulators of floral organ development, including lemmas, in barley” (lines 426-432). Although *HvMADS1* is highly expressed in developing carpels, the observed no changes in carpel phenotype between WT and mutant implied that the function of *HvMADS1* on the carpel may be redundant with other genes, so we have also discussed the need for further analysis of higher-order mutants of the *HvMADS1* gene and other barley MADS-box genes (lines 376-379).

For endosperm: we have added grain size phenotype of the cross between WT and mutant in Supplementary Fig. 3, and revised the Results part as “In addition, to rule out the effect of endosperm on grain size we crossed *mads1* (♀) with WT (♂), and found that grain size of all F2 progenies is similar to that of WT (Supplementary Fig. 3c, d) (lines 171-173).

In addition, we have discussed this issue, as suggested, in the Discussion section as “Generally, cereal grain size is mainly determined by maternal spikelet hull, pericarp, overall pistil (including carpel), and endosperm. In this study, we also demonstrated that changes in *mads1* grain weight is highly associated with changes in lemma width

(Fig. 1i, 2e). We also ruled out the carpel and endosperm as the cause of the smaller *mads1* grains (Supplementary Fig. 3a, b, c, d). Thus, the 33% reduction in bulk grain weight of *mads1* grain (Fig. 2e) is thus mainly due to the narrower lemma width; nevertheless, currently we cannot completely rule out the effect of pericarp” (lines 403-410). Notably, although the expression of *HvMADS1* in developing caryopses was comparable to that of early developing inflorescences (W2.0-W4.5), trends of them were different. The expression of *HvMADS1* increased as inflorescence developed but decreased as grain developed (Supplementary Fig. 4a), implying likely minor roles in caryopses development.

Comment 4: They state that *HvMADS1* controls awn and lemma development by directly modulating *HvSHI* and *HvDL*. Their gene-edited lines strongly support that *HvDL* promotes awn elongation and lemma lateral growth, and it was already known from loss of *HvSHI* function *lks1* alleles (Yuo et al, 2012) that *HvSHI* promotes awn elongation, and this paper provides robust support that *HvMADS1* directly binds to these target genes. These together are suggestive that the binding relationship is important for *HvMADS1* control of awn length and lemma width. However, without genetic analyses to demonstrate how this relationship impacts the phenotypes of double mutants (eg. wrt epistasis), or transgenic rescue by overexpression of *HvDL* or *HvSHI* in the *hvmads1* mutants, we do not have an estimate of the importance of this binding (or loss thereof) to the phenotypes observed in *hvmads1*. So, I would tone this conclusion down.

Answer: We agreed with your comment and thank you so much for pointing this out. We have toned down the conclusion that *HvMADS1* controls awn and lemma development by directly regulating *HvSHI* and *HvDL*.

We have revised our manuscript as suggested as shown in below.

- 1) We have revised the sentence in the Abstract “We defined two direct targets of *HvMADS1* regulation, *HvSHI* and *HvDL*” to “We defined two potential direct targets of *HvMADS1* regulation, *HvSHI* and *HvDL*” (lines 30-31).

- 2) We have revised the subtitle of this part “HvMADS1 directly modulates downstream *HvSHI* and *HvDL* to control awn size and lemma transversal growth” in the Results section as “HvMADS1 directly modulates the expression of potential downstream genes *HvSHI* and *HvDL* to control awn size and lemma transversal growth” (lines 244-245).
- 3) We have revised this sentence “These results indicated that HvMADS1 directly targets *HvSHI* and *HvDL* to regulate awn and/or lemma development through controlling cell proliferation” in the Results section as “These results implied that HvMADS1 directly targets *HvSHI* and *HvDL* to regulate awn and/or lemma development through controlling cell proliferation” (277-278).
- 4) We have added two sentences in the Discussion part as “Combining a series of biochemical experiments with the phenotypes of single mutants, we inferred that the binding of HvMADS1 to *HvSHI* and *HvDL* is critical for HvMADS1 in controlling awn length and lemma width. The future work should focus on the genetic analysis of double mutants (*mads1 shi* and *mads1 dl*), *mads1* mutants overexpressing *HvDL* and/or *HvSHI*, and transgenic plants with site-directed mutated motifs of *HvDL/HvSHI* that bound by HvMADS1, to further strengthen the understanding of the biological significance of the HvMADS1-HvSHI/HvDL module” (lines 432-438).

Comment 5: I couldn't find detailed information about the exact gene edits for the HvMADS1, HvDL, HvSHI and HvAP2 alleles described in the paper, including the changes within each allele, the screening, generation analysed or Cas9 presence/absence. It is also unclear if the *hvmads1* alleles used in this paper are the same ones described in their previous publication (Li et al., 2021) – potentially *hvmads1-5* is?

Answer: Thanks for the comment.

We have provided detailed information on the exact gene editing of the *HvMADS1*,

HvSHI, *HvDL*, and *HvAP2* alleles in Supplementary Fig. 1a, 6b, 7b, and 11b, respectively. We have also added a sentence in the M&M section as “Homozygous progeny plants of at least T₂ or higher were used in the analyses” (line 489) to clarify the generations of these mutants used in this analysis. However, we didn't test for the presence or absence of Cas9 in these mutants.

In addition, we have clarified that the *hvmads1* allele used in this paper is not the same as described in previous publication (Li et al., 2021) in the Results section as “We generated three knockout lines of *HvMADS1* using CRISPR-Cas9 system, namely *mads1-3*, *mads1-5* and *mads1-8* in barley cv. Golden Promise (Supplementary Fig. 1a), with different alleles from those reported in previous studies” (lines 123-125).

Comment 6: More explanation of the awn removal experiment is needed. When/ how were the spikes de-awned and how long after were the photosynthetic rates determined? Also, were grain from these spikes compared with awned spikes in either genotype?

Answer: Thanks for this constructive comment.

We have added in the M&M section the explanation regarding the photosynthetic rate determination and grain weight measurement as “Barley spike photosynthetic gas exchange parameters were measured as described. Briefly, the entire awn on all spikelets of uniform main spikes of healthy WT plants at heading were cut off using sharp blades, and the whole-spike net photosynthetic rate was measured three days after de-awning to rule out the effect of wounding. The grain weight of both WT and de-awned WT was measured at harvest time” (lines 495-500).

Additionally, we have not measured the grain weights of de-awned and non-de-awned *mads1* mutant, therefore, we have only added grain weight information of de-awned and non-de-awned WT in the Result section as “Most interestingly, the grain weight of de-awned WT was about 6% lower than that of the non-de-awned WT (Supplementary Fig. 1b)” (lines 140-141).

Comment 7: Their conclusion that HvMADS1 promotes growth through cell proliferation is primarily based on ‘cell number’ estimates extrapolated from the measurements of cell size within a small section of the awn and lemma. They do not discuss the possibility that cell size might change throughout the organ which could also contribute to the awn length and lemma width differences – so I would use caution the interpretation to extrapolate across the entire organ and more directly state that the cell number differences were extrapolated from the measurements of cell size within a small section of the awn and lemma.

Answer: Thank you very much for raising this point. Sorry that we did not state clearly the methodology in the previous version.

For the awn, because the awn is long and thin, it is too difficult to cut the whole awn longitudinally. Therefore, we can only count the cell length by cutting the middle part of the awn longitudinally and calculating the cell number based on the entire awn length.

We have revised this part in the M&M section as “For the length and number of awn cells, the awn was longitudinally cut at the middle part of the awn, cell length per awn across the entire cut region of the awn was measured, and the resulting value was divided by the awn length to yield cell numbers per awn” (lines 509-512).

We have also revised the information regarding the measurement of the cell size and cell number in lemma cells in the M&M section as “For the size and number of lemma cells, lemma was cut transversally at the middle part of the lemma, and cell length and cell numbers were directly measured and counted, respectively” (lines 512-514). Therefore, the cell number of the lemma was not obtained by estimation.

Comment 8: The last sentence in their discussion suggests that ‘other superior alleles of HvMADS1, natural or gene-edited, will be a key resource in the future for high-yield barley breeding’. Their previous study (Li et al., 2021) showed high levels of

conservation in HvMADS1 across barley, so it is unclear how likely natural superior alleles would be found.

Answer: Thank you for this important comment.

Li's conclusion regarding high levels of conservation in *HvMADS1* across barley were drawn from the coding region of *HvMADS1*. Therefore, we hoped to find some variations in the promoter or the first intron region of *HvMADS1*. Indeed, we analysed natural variations in *HvMADS1* using whole-genome shotgun data from 200 domesticated and 100 wild varieties of barley⁵, and detected in total 559 SNPs for *HvMADS1* (Figure 3a below); the population structure of these cultivars showed that they have different genetic backgrounds (Figure 3b below). However, we did not find the association of SNPs with phenotype, further research is needed to determine if there are natural superior alleles.

As you suggested, we have revised this part as “so genome-edited superior alleles of MADS1 will be a key resource in the future for high-yield barley breeding.” (lines 459-461).

Figure 3. SNPs in *HvMADS1* and the population structure of 300 barley varieties

Comment 9: Introduction incorrectly states that SHORT INTERNODES is the only gene reported in barley to control awn elongation. Shoosmith et al (2021) demonstrated that loss of APETALA2 function leads to shortened lemma awns.

Answer: Thank you for pointing out the mistake.

We have revised this part in the Introduction section as “In barley, *SHORT INTERNODES (SHI)* controls awn elongation and pistil morphology and *APETALA2 (AP2)* regulates awn development and awn-lemma boundary” (lines 85-87).

Comment 10: I’m not convinced that the H4 in situ shows a gradual attenuation of signal, although the awn itself is obviously smaller.

Answer: Thank you for this comment.

Although we confirmed that the expression of cell cycle-related genes is generally down-regulated in the lemma and awn of *mads1* by RT-qPCR (Supplementary Fig. 5 c), to avoid confusion, we have deleted H4 related results in the revised version.

Comment 11: Could the increase in Shi::LUC expression when AP2 and MADS1 are co-expressed be explained by an additive effect? i.e. both AP2 and MADS1 binding separate reporter constructs – unless you think the proteins are provided in excess to the availability of the reporter? Something to consider.

Answer: Thanks for this insightful comment.

Our current results can't rule out the possibility you raised. However, Considering the interaction between AP2 and MADS1 interactions (Fig.6a-e), this interaction can enhance the transcriptional activation of MADS1 (Fig. 6f), and genetics data (Fig. 6l), we believe that HvMADS1 synergistically regulates downstream genes with HvAP2 (lines 305-314).

Comment 12: Figure 3b far right-hand graph in panel describing the outer parenchyma cell number – was this determined from the SEM or a tissue section. This wasn't clear.

Answer: Thank you for this comment, we are sorry to have missed this information.

We have revised the legends of Fig. 3 of this part as “At W4.5, W5.5 and W6.5 stages, awns were photographed using a stereoscope and their lengths were calculated using Image J. At other stages, the awn length was measured directly with a ruler. The width and cell number of lemma were obtained by analyzing lemma sections ($n \geq 18$ individual spiklets for awn length and $n \geq 9$ individual lemma samples for lemma width and cell number)” (lines 934-938). In the M&M, we have also added detailed methodology for such analysis (Lines 509-514).

Comment 12: Figure 5a and 5j – I’m not clear what the RNAseq panel is showing – does this plot represent reads from either genotype indicating a loss of reads in the mutant? Also, both panels are too small to read easily.

Answer: Thank you for this comment. We have increased the size of panel a and panel j in the revised Fig. 5.

This RNA-Seq panel indicates that numbers of reads enriched in *SHI* (a) or *DL* (j) in *mads1-8* is significantly less than that in WT, proving that expression of *SHI* (a) or *DL* (j) is downregulated in mutant.

Comment 13: Figure 5n ‘M’ from ‘MADS1’ is missing as an x-axis label. Furthermore, the figure caption does not describe co-expressed AP2 being part of this experiment.

Answer: Thank you very much for your comment, we apologized for the mistakes and have corrected it in this revised manuscript as “HvMADS1 and HvMADS1-HvAP2 interaction activate expression of promoters of *HvSHI* (e) and *HvDL* (n) in barley protoplasts” (line 973-975).

Response to comments of Reviewer #3:

Overall comment: Zhang et al. study the effect of knockout mutations of barley E-class homeobox gene HvMADS1 on awn, lemma and seed development, inspired by the effects of similar analyses in rice. They could demonstrate that HvMADS1 is involved in awn/lemma development, likely through the interaction with HvSHI and

HvAP2. The work builds on previous studies and takes advantage of a number of molecular biology tools, e.g. MADS1-GFP fusions which were required for the cut&tag analysis. The effect on awn length was reproduced for triple KO mutants of the putative orthologous genes in wheat. Overall the study looks convincing and due to the similarities but also contrasts between rice, barley and wheat it is likely that the study will be interesting to a broader molecular plant biology/developmental biology audience.

Answer: Thank you so much for the encouragement and your constructive comments.

Comment 1: My main concerns are about the yield claims. I think authors should be very careful with making strong yield claims without documenting proper conclusive field experiments. There is no information on the size and the design of the field experiments hence the statistical analysis provided for yield parameters is non-conclusive.

Answer: Thank you very much for raising this critical issue.

To address your concerns, we have made following changes.

(1) We have revised relevant part in M&M to provide information on the size and the design of the field experiments as “For field experiments, each WT and mads1 mutant barley line was planted in ten rows with 0.20 m between rows, the length of each row is about 1 meter and 15 seeds were planted per row. These barley plants were grown under normal conditions and followed local agricultural practices for field management, avoiding floods, droughts, pests and diseases. 300 kg compound fertilizer per hectare (Nitrogen: Phosphorus: Potassium = 15%: 15%: 15%.) and 250 kg urea fertilizer per hectare were applied at the sowing and jointing stages, respectively. In the June of the following year, plants were randomly sampled to measure spike length, spikelet number per spike/plant, grain number per spike/plant, grain weight per plant, seed setting and thousand-grain weight. Grain weight and size were measured in grains that were treated in a 37°C oven for one week.” (lines 467-477).

- (2) We have added field data for 2022/2023 in the Results section as “The weight of *mads1* grains was about 32.7% lower than that of WT grain (Fig. 2e). This agronomically important result was confirmed in two-year field trials (Supplementary Fig. 2c)” (lines 176-178).
- (3) We have also discussed field trial-related content in the Discussion section as “It is worth noting that plot sizes used in our field trials (around 2 m²/line) were much smaller than those typically used by breeders in breeding selections” (lines 410-412).
- (4) We have added additional sentence in the Discussion section to weaken our claim as “In addition, our field experiments were carried out in the same region for two consecutive years without setting up of different plot replications for the same line. Therefore, larger-scale field trials with plot replications in different regions are needed for subsequent field yield analyses” (lines 412-415).

Comment 2: Methods are too brief for other sections, e.g. no information is available on sample size for all morphological and cell biology experiments. In the molecular experiments the definition of replicates needs to be explained and defined. To my opinion the authors have worked exclusively with technical controls.

Answer: Thank you so much for pointing out this problem, and we apologize for not giving details of the sample size and the number of replications of the experiment in our methods section.

Indeed, the sample size was labeled in each figure, denoted by “n=xx”, and the number of repetitions were also given in the figure legends.

However, as suggested, we have added detailed description in M&M section (lines 466-467; 492-500, 508-514, 528-529, 546) and in Figure legend (lines 899-910, 920-925, 937-938, 942-943, 986, 1010, 1012-1013, 1021, 1047, 1056-1057, 1067, 1075, 1098, 1107-1108, 1122, 1129-1130, 1136-1137, 1144-1145, 1158, 1162, 1173) in the revised manuscript. We have also included the Statistical analyses section in the revised

manuscript (lines 635-643).

Comment 3: data availability: the authors say that "Source data are provided with this paper. Additional data supporting the findings of this study are available from the corresponding author upon reasonable request". This is basically not true and the manuscript must not be accepted without free access to all sequencing raw data which must be submitted to public repositories. Plant material should be provided with a proper Biosample ID. The notion "available upon reasonable request" should never be accepted because every published result must be open to reproduction, therefore materials must be made available without any restriction.

Answer: Thank you very much for your comment, we apologized for missing this. We have uploaded transcriptome and CUT&TAG data to NCBI and provided accession number in the Data availability section as "The transcriptomic data and CUT&Tag data generated in this paper have been deposited in the National Center for Biotechnology Information under accession number GSE228410 and PRJNA1012547, respectively. Requests for plant materials should be addressed to JX.S. Source data are provided with this paper" (lines 647-651).

Cited references

1. Theißen G, Melzer R, Rümpler F. MADS-domain transcription factors and the floral quartet model of flower development: Linking plant development and evolution. *Development*, 2016, 143: 3259-3271
2. Hu Y, Liang W, Yin C, et al. Interactions of *OsMADS1* with floral homeotic genes in rice flower development. *Mol Plant*, 2015, 8: 1366-1384
3. Shoesmith JR, Solomon CU, Yang X, et al. APETALA2 functions as a temporal factor together with BLADE-ON-PETIOLE2 and MADS29 to control flower and grain development in barley. *Development*, 2021, 148
4. Krogan NT, Hogan K, Long JA. APETALA2 negatively regulates multiple floral organ identity genes in Arabidopsis by recruiting the co-repressor TOPLESS and the histone deacetylase HDA19. *Development*, 2012, 139: 4180-4190.

REVIEWER COMMENTS

Reviewer #1 (Remarks to the Author):

The authors have addressed my concerns appropriately in their response letter. I do not see a tracked changes version of the manuscript in the available files?

Reviewer #2 (Remarks to the Author):

Reviewer Summary: In their revision, Zhang and co-authors present a strengthened manuscript describing their functional characterization of MADS1 in barley floral development and its crucial role in awn and lemma formation. They extend their previous work to investigate possible targets and co-factors involved in HvMADS1 function using powerful and appropriate genetic and biochemical experiments. The revised paper provides the missing methodological details and for the most part addresses my comments from the original draft. For ease of understanding, I address the author responses to my comments, indicating the few instances where I suggest further revision or rewording. Lastly, I am very saddened to learn of the tragic passing of Prof Dabing Zhang.

Response to comments of Reviewer #2:

Overall comment: Zhang et al's (2023) well-crafted manuscript describes the function of the barley transcription factor-encoding gene HvMADS1. This paper builds on their previous work showing that loss of HvMADS1 function causes changes in reproductive architecture (Li et al., 2021). Here they focus on HvMADS1' control of awn and lemma growth. They demonstrate that *hvmads1* mutants have shorter awns and narrower lemmas, which in both cases may result from reduced cell number, suggesting that HvMADS1 promotes cell proliferation during organ development. They then associate these differences with reductions in spike photosynthetic rate as well as reduced grain width, depth and weight in the mutant. In a significant advance, they use comparative transcriptomics and CUT&Tag assays to reveal multiple direct genomic targets of HvMADS1, including HvDROOPING LEAF (HvDL) and HvSHORTENED INTERNODES, genes previously shown to promote awn development in rice and barley, respectively. They test the functional relevance of these targets by generating and characterising loss of function *hvd1* and *hvshi* alleles which phenocopied multiple aspects of the *hvmads1* phenotype, suggesting that HvMADS1 works through these targets to control lemma and awn development. In addition, they show that HvMADS1 interacts with HvAPETALA2, a factor which regulates similar traits, and that HvMADS1's transcriptional activation may increase in the presence of HvAP2. Lastly, gene editing MADS1 in wheat suggests that MADS1's role in controlling awn length are conserved between these species. Taken together, they reveal that how HvMADS1 may work as an upstream regulator of a pathway controlling awn and lemma development which is correlated with final grain parameters and yield in barley.

This paper's combination of molecular and functional insight about the genes important for lemma and awn growth provides a major advance in our understanding of development of these grass-specific floral organs in temperate cereals, especially with regards to the targeting of HvDL and HvSHI by HvMADS1, as the upstream regulation of these genes is not well understood in grass models.

Answer: We deeply appreciated your positive comments and great support on our manuscript.

Comment 1: However, some results in this paper overlap with existing data and some ideas published in Shoesmith et al (2021), including the HvMADS1 in situ as well as the floral and grain phenotypes due to a loss of HvAP2 function, which could have been better cited with reference to their own results.

Answer: We appreciated very much for this comment.

We have revised and cited relevant references as suggested. For HvAP2, we have cited relevant article in the Introduction section “In barley, SHORT INTERNODES (SHI) controls awn elongation and pistil morphology and APETALA2 (AP2) regulates awn development and awn-lemma boundary” (lines 85-87) and in the Discussion section “Previous studies have shown that HvAP2 has pleiotropic roles in floral organ and grain development, regulating lemma longitudinal and pericarp transversal development and thus grain length and width” (lines 439-441). In addition, we have revised the manuscript in the Results section regarding awn and grain size of ap2 mutants as “The observed phenotypes of awn length and grain size of ap2 mutants are consistent with previous reports” (lines 305-307).

Reviewer comment: Thank you to the authors for citing the previous work on HvAP2 appropriately. However, could the authors please revise lines 85-87 to mention that HvAP2 controls lemma identity. Also please revise (lines 439-441) “regulating lemma longitudinal and pericarp transversal development and thus grain length and width” to “regulating lemma identity and length as well as grain length and width”. I’m also unsure that the use of ‘transversal’ as an adjective is correct; transversal means when a line crosses two other lines. I suggest that the authors use another term to describe the transverse direction throughout the paper.

For HvMADS1 short awn phenotype, we described short awn phenotype in our mutants and cited previously reported as “All three homozygous mutant plants also exhibited significantly shorter awns than WT plants (Fig. 1a, d), confirming an important role of HvMADS1 in awn development as previously reported” (lines 125-128). For HvMADS1 in situ hybridization, Shoesmith’s work showed that the expression of MADS1 in young and old spikelets particularly in developing glumes and lemma/palea of the Bowman background is relative weak; here, we provided a clear and strong in situ expression pattern of MADS1 in the developing inflorescence of Golden Promise background (Fig. 3c).

Comment 2: Furthermore, this paper repeats some information from their previous manuscript Li et al (2021) which showed both the short awns and “retarded awn elongation in the mutant (Extended Data Fig. 5a,b)”, although the SEM images in this manuscript are different.

Answer: We appreciated very much for this suggestive comment to improve our manuscript quality.

We have revised this part thoroughly and clarified that the HvMADS1 alleles used in this study is different from previously reported one (Li et al., 2021) in the Results section as “We generated three knockout lines of HvMADS1 using CRISPR-Cas9 system, namely mads1-3, mads1-5 and mads1-8 in barley cv. Golden Promise (Supplementary Fig. 1a), with different alleles from those reported in previous studies. All three homozygous mutant plants also exhibited significantly shorter awns than WT plants (Fig. 1a, d), confirming an important role of HvMADS1 in awn development as previously reported” (lines 123-128).

Reviewer comment: I agree with the authors that the HvMADS1 in situ in Golden Promise shows a stronger signal compared to Bowman as reported in Shoesmith et al., (2021).

However, Shoesmith et al. (2021) also described that the gain of HvAP2 function mutant showed the strong spatial and quantitative upregulation of HvMADS1 (by qPCR and in situ) which supported the interpretation that HvMADS1 may contribute to HvAP2-mediated changes in awn development. These findings and ideas are relevant and similar to those reported in this paper – i.e. that HvAP2 and HvMADS1 may work synergistically to control awn development. This manuscript would be improved by integrating this previous work which supports their argument, along with appropriate citations.

Therefore, although our work was an extension of Li's to reveal underlying molecular mechanisms, the HvMADS1 alleles used in our study is different from that used by Li. In addition, Li's work was carried out in an Australian laboratory, ours was done independently in a Chinese laboratory.

Reviewer comment: Thank you for the clarification.

Comment 3: One major concern is the manuscript's message that grain differences due to a loss of HvMADS1 function are caused by changes in lemma/ awn growth. At multiple points in the manuscript, grain changes are described as being caused by the changes in lemma development (eg. line 26 "HvMADS1 positively regulates awn length and lemma width, leading to increased grain size and weight"). However, they only show a correlation and do not provide evidence for a causative relationship, This is also a concern with the research in rice cited in their paper. Alternative or additional possibilities should be addressed or considered, such as a role for HvMADS1 in the carpel, caryopsis or grain which could contribute to the grain differences, especially as HvMADS1 is relatively highly expressed in developing carpels and caryopses. Furthermore, the putative direct target of MADS1, HvDL clearly plays a major role in the carpel according to their manuscript (line 250-251 "hvd1 mutants were unable to set seeds due to defective carpels"), consistent with its well-known role in rice to specify the carpel (Yamaguchi et al., 2004). Thus, both HvMADS1 and HvDL could influence carpel characteristics.

Answer: This is a wonderful comment that helps us a lot to clarify the function of HvMADS1 in grain size.

To address your concerns, we have performed additional analyses, provided more results, and revised the corresponding parts.

For carpel: 1) we have compared the carpel size between WT and mads1 mutant, added the results in Supplementary Fig. 3, and revised the Results part as "To rule out the effect of carpel size on grain size, we also compared carpels size between WT and mads1 mutant, and found no significant difference in the carpels size between WT and mutant at W10.0 stage (Supplementary Fig. 3a, b)" (lines 168-171); 2) We also added the carpel phenotype of dl mutant in Supplementary Fig. 7h, and revised corresponding part in the Discussion part as "Interestingly, HvDL also controls carpel properties (Supplementary Fig. 7h) and the number of lemma vascular bundles (Supplementary Fig. 10i), suggesting that there may be other proteins or pathways that control HvDL expression to regulate floral organ properties. Further investigations into upstream and downstream regulatory networks of the HvMADS1–DL module may help to identify the molecular regulators of floral organ development, including lemmas, in barley" (lines 426-432). Although HvMADS1 is highly expressed in developing carpels, the observed no changes in carpel phenotype between WT and mutant implied that the function of HvMADS1 on the carpel may be reductant with other genes, so we have also discussed the need for further analysis of higher-order mutants of the HvMADS1 gene and other barley MADS-box genes (lines 376-379).

For endosperm: we have added grain size phenotype of the cross between WT and mutant in Supplementary Fig. 3, and revised the Results part as “In addition, to rule out the effect of endosperm on grain size we crossed *mads1* (♀) with WT (♂), and found that grain size of all F2 progenies is similar to that of WT (Supplementary Fig. 3c, d) (lines 171-173).

In addition, we have discussed this issue, as suggested, in the Discussion section as “Generally, cereal grain size is mainly determined by maternal spikelet hull, pericarp, overall pistil (including carpel), and endosperm. In this study, we also demonstrated that changes in *mads1* grain weight is highly associated with changes in lemma width (Fig. 1i, 2e). We also ruled out the carpel and endosperm as the cause of the smaller *mads1* grains (Supplementary Fig. 3a, b, c, d). Thus, the 33% reduction in bulk grain weight of *mads1* grain (Fig. 2e) is thus mainly due to the narrower lemma width; nevertheless, currently we cannot completely rule out the effect of pericarp” (lines 403-410). Notably, although the expression of HvMADS1 in developing caryopses was comparable to that of early developing inflorescences (W2.0-W4.5), trends of them were different. The expression of HvMADS1 increased as inflorescence developed but decreased as grain developed (Supplementary Fig. 4a), implying likely minor roles in caryopses development.

Reviewer comment: Thanks to the authors for this further work. The data suggesting no effect on *hvmads1* on the measured carpel parameters is helpful. The F2 seed analyses is also very insightful about possible endosperm effects. The reworded discussion addresses my concerns in the original draft.

Comment 4: They state that HvMADS1 controls awn and lemma development by directly modulating HvSHI and HvDL. Their gene-edited lines strongly support that HvDL promotes awn elongation and lemma lateral growth, and it was already known from loss of HvSHI function *lks1* alleles (Yuo et al, 2012) that HvSHI promotes awn elongation, and this paper provides robust support that HvMADS1 directly binds to these target genes. These together are suggestive that the binding relationship is important for HvMADS1 control of awn length and lemma width. However, without genetic analyses to demonstrate how this relationship impacts the phenotypes of double mutants (eg. wrt epistasis), or transgenic rescue by overexpression of HvDL or HvSHI in the *hvmads1* mutants, we do not have an estimate of the importance of this binding (or loss thereof) to the phenotypes observed in *hvmads1*. So, I would tone this conclusion down.

Answer: We agreed with your comment and thank you so much for pointing this out. We have toned down the conclusion that HvMADS1 controls awn and lemma development by directly regulating HvSHI and HvDL.

We have revised our manuscript as suggested as shown in below.

1) We have revised the sentence in the Abstract “We defined two direct targets of HvMADS1 regulation, HvSHI and HvDL” to “We defined two potential direct targets of HvMADS1 regulation, HvSHI and HvDL” (lines 30-31).

2) We have revised the subtitle of this part “HvMADS1 directly modulates downstream HvSHI and HvDL to control awn size and lemma transversal growth” in the Results section as “HvMADS1 directly modulates the expression of potential downstream genes HvSHI and HvDL to control awn size and lemma transversal growth” (lines 244-245).

3) We have revised this sentence “These results indicated that HvMADS1 directly targets HvSHI and HvDL to regulate awn and/or lemma development through controlling cell proliferation” in the Results section as “These results implied that HvMADS1 directly targets HvSHI and HvDL to regulate awn and/or lemma development through controlling cell proliferation” (277-278).

4) We have added two sentences in the Discussion part as “Combining a series of biochemical experiments with the phenotypes of single mutants, we inferred that the binding of HvMADS1 to HvSHI and HvDL is critical for HvMADS1 in controlling awn length and lemma width. The future work should focus on the genetic analysis of double mutants (mads1 shi and mads1 dl), mads1 mutants overexpressing HvDL and/or HvSHI, and transgenic plants with site-directed mutated motifs of HvDL/HvSHI that bound by HvMADS1, to further strengthen the understanding of the biological significance of the HvMADS1-HvSHI/HvDL module” (lines 432-438).

Reviewer comment: Thank you for the revisions. However, the revised subtitle text described in point 2 still states that HvMADS1 regulation of DL and SHI causes the changes in lemma and awn. My concern was not the confidence about whether HvMADS1 binds these targets or not, but whether this activity is directly responsible for the changes in lemma development in the hvmads1 mutants. Similarly, I suggest that Point 3 still over-reaches. Consider rephrasing along the lines of: “HvMADS1 could regulate awn and lemma development through direct regulation of potential targets HvSHI and HvDL” and/or “HvMADS1 direct regulation of HvSHI and HvDL could contribute to HvMADS1-dependent regulation awn and/or lemma development through controlling cell proliferation”. The changes to the discussion text described in point 4 work very well to suggest caveats and future experiments.

Comment 5: I couldn't find detailed information about the exact gene edits for the HvMADS1, HvDL, HvSHI and HvAP2 alleles described in the paper, including the changes within each allele, the screening, generation analysed or Cas9 presence/absence. It is also unclear if the hvmads1 alleles used in this paper are the same ones described in their previous publication (Li et al., 2021) – potentially hvmads1-5 is?

Answer: Thanks for the comment.

We have provided detailed information on the exact gene editing of the HvMADS1, HvSHI, HvDL, and HvAP2 alleles in Supplementary Fig. 1a, 6b, 7b, and 11b, respectively. We have also added a sentence in the M&M section as “Homozygous progeny plants of at least T2 or higher were used in the analyses” (line 489) to clarify the generations of these mutants used in this analysis. However, we didn't test for the presence or absence of Cas9 in these mutants.

In addition, we have clarified that the hvmads1 allele used in this paper is not the same as described in previous publication (Li et al., 2021) in the Results section as “We generated three knockout lines of HvMADS1 using CRISPR-Cas9 system, namely mads1-3, mads1-5 and mads1-8 in barley cv. Golden Promise (Supplementary Fig. 1a), with different alleles from those reported in previous studies” (lines 123-125).

Reviewer comment: Thank you for providing all these details and clarifications. It is unfortunate that these lines were not examined for presence of the Cas9 insert. From what I see presented in Supplemental Figure 6c-e only one of the two shi alleles were characterized. Without showing that these mutants are Cas9-free, you cannot rule out the possibility that the Cas9 insert itself is causing these phenotypes and this needs to be addressed. For mads1, ap2 and dl, multiple alleles show the same phenotype, which makes contributions from the Cas9 insert to the phenotype unlikely.

Comment 6: More explanation of the awn removal experiment is needed. When/ how were the spikes de-awned and how long after were the photosynthetic rates determined? Also,

were grain from these spikes compared with awned spikes in either genotype?

Answer: Thanks for this constructive comment.

We have added in the M&M section the explanation regarding the photosynthetic rate determination and grain weight measurement as “Barley spike photosynthetic gas exchange parameters were measured as described. Briefly, the entire awn on all spikelets of uniform main spikes of healthy WT plants at heading were cut off using sharp blades, and the whole-spike net photosynthetic rate was measured three days after de-awning to rule out the effect of wounding. The grain weight of both WT and de-awned WT was measured at harvest time” (lines 495-500).

Additionally, we have not measured the grain weights of de-awned and non-de-awned *mads1* mutant, therefore, we have only added grain weight information of de-awned and non-de-awned WT in the Result section as “Most interestingly, the grain weight of de-awned WT was about 6% lower than that of the non-de-awned WT (Supplementary Fig. 1b)” (lines 140-141).

Reviewer comment: Thank you for providing all these methodological details

Comment 7: Their conclusion that *HvMADS1* promotes growth through cell proliferation is primarily based on ‘cell number’ estimates extrapolated from the measurements of cell size within a small section of the awn and lemma. They do not discuss the possibility that cell size might change throughout the organ which could also contribute to the awn length and lemma width differences – so I would use caution the interpretation to extrapolate across the entire organ and more directly state that the cell number differences were extrapolated from the measurements of cell size within a small section of the awn and lemma.

Answer: Thank you very much for raising this point. Sorry that we did not state clearly the methodology in the previous version.

For the awn, because the awn is long and thin, it is too difficult to cut the whole awn longitudinally. Therefore, we can only count the cell length by cutting the middle part of the awn longitudinally and calculating the cell number based on the entire awn length.

We have revised this part in the M&M section as “For the length and number of awn cells, the awn was longitudinally cut at the middle part of the awn, cell length per awn across the entire cut region of the awn was measured, and the resulting value was divided by the awn length to yield cell numbers per awn” (lines 509-512).

We have also revised the information regarding the measurement of the cell size and cell number in lemma cells in the M&M section as “For the size and number of lemma cells, lemma was cut transversally at the middle part of the lemma, and cell length and cell numbers were directly measured and counted, respectively” (lines 512-514). Therefore, the cell number of the lemma was not obtained by estimation.

Reviewer comment: Thank you for the clarification that cell lengths across the entire awn were measured and how the lemma cell number and sizes were measured.

Comment 8: The last sentence in their discussion suggests that ‘other superior alleles of *HvMADS1*, natural or gene-edited, will be a key resource in the future for high-yield barley

breeding". Their previous study (Li et al., 2021) showed high levels of conservation in HvMADS1 across barley, so it is unclear how likely natural superior alleles would be found.
Answer: Thank you for this important comment.

Li's conclusion regarding high levels of conservation in HvMADS1 across barley were drawn from the coding region of HvMADS1. Therefore, we hoped to find some variations in the promoter or the first intron region of HvMADS1. Indeed, we analysed natural variations in HvMADS1 using whole-genome shotgun data from 200 domesticated and 100 wild varieties of barley⁵, and detected in total 559 SNPs for HvMADS1 (Figure 3a below); the population structure of these cultivars showed that they have different genetic backgrounds (Figure 3b below). However, we did not find the association of SNPs with phenotype, further research is needed to determine if there are natural superior alleles.

As you suggested, we have revised this part as "so genome-edited superior alleles of MADS1 will be a key resource in the future for high-yield barley breeding." (lines 459-461).

Figure 3. SNPs in HvMADS1 and the population structure of 300 barley varieties

Reviewer comment: This revision works well.

Comment 9: Introduction incorrectly states that SHORT INTERNODES is the only gene reported in barley to control awn elongation. Shoemith et al (2021) demonstrated that loss of APETALA2 function leads to shortened lemma awns.

Answer: Thank you for pointing out the mistake.

We have revised this part in the Introduction section as "In barley, SHORT INTERNODES (SHI) controls awn elongation and pistil morphology and APETALA2 (AP2) regulates awn development and awn-lemma boundary" (lines 85-87).

Reviewer comment: Comment addressed, thank you.

Comment 10: I'm not convinced that the H4 in situ shows a gradual attenuation of signal, although the awn itself is obviously smaller.

Answer: Thank you for this comment.

Although we confirmed that the expression of cell cycle-related genes is generally down-regulated in the lemma and awn of *mads1* by RT-qPCR (Supplementary Fig. 5 c), to avoid confusion, we have deleted H4 related results in the revised version.

Reviewer comment: Concern addressed, thank you.

Comment 11: Could the increase in Shi::LUC expression when AP2 and MADS1 are co-expressed be explained by an additive effect? i.e. both AP2 and MADS1 binding separate reporter constructs – unless you think the proteins are provided in excess to the availability of the reporter? Something to consider.

Answer: Thanks for this insightful comment.

Our current results can't rule out the possibility you raised. However, Considering the interaction between AP2 and MADS1 interactions (Fig.6a-e), this interaction can enhance the transcriptional activation of MADS1 (Fig. 6f), and genetics data (Fig. 6l), we believe that HvMADS1 synergistically regulates downstream genes with HvAP2 (lines 305-314).

Comment 12: Figure 3b far right-hand graph in panel describing the outer parenchyma cell number – was this determined from the SEM or a tissue section. This wasn't clear.

Answer: Thank you for this comment, we are sorry to have missed this information.

We have revised the legends of Fig. 3 of this part as “At W4.5, W5.5 and W6.5 stages, awns were photographed using a stereoscope and their lengths were calculated using Image J. At other stages, the awn length was measured directly with a ruler. The width and cell number of lemma were obtained by analyzing lemma sections ($n \geq 18$ individual spiklets for awn length and $n \geq 9$ individual lemma samples for lemma width and cell number)” (lines 934-938). In the M&M, we have also added detailed methodology for such analysis (Lines 509-514).

Reviewer comment: Comment addressed, thank you.

Comment 12: Figure 5a and 5j – I'm not clear what the RNAseq panel is showing – does this plot represent reads from either genotype indicating a loss of reads in the mutant? Also, both panels are too small to read easily.

Answer: Thank you for this comment. We have increased the size of panel a and panel j in the revised Fig. 5.

This RNA-Seq panel indicates that numbers of reads enriched in SHI (a) or DL (j) in mads1-8 is significantly less than that in WT, proving that expression of SHI (a) or DL (j) is downregulated in mutant.

Reviewer comment: Comment addressed, thank you.

Comment 13: Figure 5n 'M' from 'MADS1' is missing as an x-axis label. Furthermore, the figure caption does not describe co-expressed AP2 being part of this experiment.

Answer: Thank you very much for your comment, we apologized for the mistakes and have corrected it in this revised manuscript as “HvMADS1 and HvMADS1-HvAP2 interaction activate expression of promoters of HvSHI (e) and HvDL (n) in barley protoplasts” (line 973-975).

Reviewer comment: Comment addressed, thank you.

Reviewer #3 (Remarks to the Author):

The authors have addressed all my concerns appropriately. No further comments.

Point-by-point responses to the comments by Reviewers

Response to comments of Reviewer #1

Reviewer #1 (Remarks to the Author):

The authors have addressed my concerns appropriately in their response letter. I do not see a tracked changes version of the manuscript in the available files?

Answer: We appreciated very much for your great support to our work. We are sorry that you did not see a tracked changes. Perhaps, the tracked version was not included in the converted PDF file, because we have submitted both clean and tracked versions online as required.

Response to comments of Reviewer #2

Reviewer #2 (Remarks to the Author):

Reviewer Summary: In their revision, Zhang and co-authors present a strengthened manuscript describing their functional characterization of MADS1 in barley floral development and its crucial role in awn and lemma formation. They extend their previous work to investigate possible targets and co-factors involved in HvMADS1 function using powerful and appropriate genetic and biochemical experiments. The revised paper provides the missing methodological details and for the most part addresses my comments from the original draft. For ease of understanding, I address the author responses to my comments, indicating the few instances where I suggest further revision or rewording. Lastly, I am very saddened to learn of the tragic passing of Prof Dabing Zhang.

Answer: Thank you very much for your time and efforts in providing useful and constructive comments for further improvement of our manuscript. We appreciate your careful and deliberate review. We have revised the manuscript as you suggested, and we hope it will address all your concerns.

Yes, it is a heavy loss of Prof. Zhang, he will be always missed by his colleagues, friends, and students. Thank you again.

Response to comments of Reviewer #2:

Overall comment: Zhang et al's (2023) well-crafted manuscript describes the function of the barley transcription factor-encoding gene HvMADS1. This paper builds on their previous work showing that loss of HvMADS1 function causes changes in reproductive architecture (Li et al., 2021). Here they focus on HvMADS1' control of awn and lemma growth. They demonstrate that *hvmads1* mutants have shorter awns and narrower lemmas, which in both cases may result from reduced cell number, suggesting that HvMADS1 promotes cell proliferation during organ development. They then associate these differences with reductions in spike photosynthetic rate as well as reduced grain width, depth and weight in the mutant. In a significant advance, they use comparative transcriptomics and CUT&Tag assays to reveal multiple direct genomic targets of HvMADS1, including HvDROOPING LEAF (HvDL) and HvSHORTENED INTERNODES, genes previously shown to promote awn development in rice and barley, respectively. They test the functional relevance of these targets by generating and characterising loss of function *hvd1* and *hvs1* alleles which phenocopied multiple aspects of the *hvmads1* phenotype, suggesting that HvMADS1 works through these targets to control lemma and awn development. In addition, they show that HvMADS1 interacts with HvAPETALA2, a factor which regulates similar traits, and that HvMADS1's transcriptional activation may increase in the presence of HvAP2. Lastly, gene editing MADS1 in wheat suggests that MADS1's role in controlling awn length are conserved between these species. Taken together, they reveal that how HvMADS1 may work as an upstream regulator of a pathway controlling awn and lemma development which is correlated with final grain parameters and yield in barley.

This paper's combination of molecular and functional insight about the genes important for lemma and awn growth provides a major advance in our understanding of development of these grass-specific floral organs in temperate cereals, especially with regards to the targeting of HvDL and HvSHI by HvMADS1, as the upstream regulation of these genes is not well understood in grass models.

Answer: We deeply appreciated your positive comments and great support on our manuscript.

Comment 1: However, some results in this paper overlap with existing data and some ideas published in Shoesmith et al (2021), including the HvMADS1 in situ as well as the floral and grain phenotypes due to a loss of HvAP2 function, which could have been better cited with reference to their own results.

Answer: We appreciated very much for this comment.

We have revised and cited relevant references as suggested. For HvAP2, we have cited relevant article in the Introduction section “In barley, SHORT INTERNODES (SHI) controls awn elongation and pistil morphology and APETALA2 (AP2) regulates awn development and awn-lemma boundary” (lines 85-87) and in the Discussion section “Previous studies have shown that HvAP2 has pleiotropic roles in floral organ and grain development, regulating lemma longitudinal and pericarp transversal development and thus grain length and width” (lines 439-441). In addition, we have revised the manuscript in the Results section regarding awn and grain size of ap2 mutants as “The observed phenotypes of awn length and grain size of ap2 mutants are consistent with previous reports” (lines 305-307).

2nd Reviewer comment#1: Thank you to the authors for citing the previous work on HvAP2 appropriately. However, could the authors please revise lines 85-87 to mention that HvAP2 controls lemma identity. Also please revise (lines 439-441) “regulating lemma longitudinal and pericarp transversal development and thus grain length and width” to “regulating lemma identity and length as well as grain length and width”. I’m also unsure that the use of ‘transversal’ as an adjective is correct; transversal means when a line crosses two other lines. I suggest that the authors use another term to describe the transverse direction throughout the paper.

Answer: Thank you so much for the careful review.

- 1) We have revised this part as “*APETALA2 (AP2)* regulates awn development, awn-lemma boundary and lemma identity” (lines 86-87).
- 2) As you suggested, we have revised “regulating lemma longitudinal and pericarp transversal development and thus grain length and width” to “regulating lemma

identity and length as well as grain length and width” (lines 444).

- 3) As you suggested, we have replaced “transversal” to “transverse” throughout the whole manuscript (lines 105, 131, 187, 209, 323, 446, 447, 458, 905, 911, 1053).

For HvMADS1 short awn phenotype, we described short awn phenotype in our mutants and cited previously reported as “All three homozygous mutant plants also exhibited significantly shorter awns than WT plants (Fig. 1a, d), confirming an important role of HvMADS1 in awn development as previously reported” (lines 125-128). For HvMADS1 in situ hybridization, Shoesmith’s work showed that the expression of MADS1 in young and old spikelets particularly in developing glumes and lemma/palea of the Bowman background is relative weak; here, we provided a clear and strong in situ expression pattern of MADS1 in the developing inflorescence of Golden Promise background (Fig. 3c).

Comment 2: Furthermore, this paper repeats some information from their previous manuscript Li et al (2021) which showed both the short awns and “retarded awn elongation in the mutant (Extended Data Fig. 5a,b)”, although the SEM images in this manuscript are different.

Answer: We appreciated very much for this suggestive comment to improve our manuscript quality.

We have revised this part thoroughly and clarified that the HvMADS1 alleles used in this study is different from previously reported one (Li et al., 2021) in the Results section as “We generated three knockout lines of HvMADS1 using CRISPR-Cas9 system, namely mads1-3, mads1-5 and mads1-8 in barley cv. Golden Promise (Supplementary Fig. 1a), with different alleles from those reported in previous studies. All three homozygous mutant plants also exhibited significantly shorter awns than WT plants (Fig. 1a, d), confirming an important role of HvMADS1 in awn development as previously reported” (lines 123-128).

2nd Reviewer comment#2-1: I agree with the authors that the HvMADS1 in situ in Golden Promise shows a stronger signal compared to Bowman as reported in Shoesmith

et al., (2021). However, Shoesmith et al. (2021) also described that the gain of HvAP2 function mutant showed the strong spatial and quantitative upregulation of HvMADS1 (by qPCR and in situ) which supported the interpretation that HvMADS1 may contribute to HvAP2-mediated changes in awn development. These findings and ideas are relevant and similar to those reported in this paper – i.e. that HvAP2 and HvMADS1 may work synergistically to control awn development. This manuscript would be improved by integrating this previous work which supports their argument, along with appropriate citations.

Answer: Thank you for this constructive comment. Indeed, the work of Shoesmith et al., (2021) triggered our think about the roles of AP2-MADS1 module in floral organ development and grain development in barley.

First, we have cited this reference in many parts of the revised manuscript (lines 87, 288, 309, 444, 446, 450).

Second, we have compared similarities and differences between our results and theirs (Shoesmith et al., 2021) regarding the expression of HvMADS1 and function of HvMADS1-Hv AP2. (1) They reported that as compared with Bowman, the HvMADS1 probe gives strong signals in young and old *Zeol.b* spikelets, especially in developing glume, lemma/palea, lodicule and stamen primordium, indicating that the ectopic expression of HvAP2 promoted the expression of HvMADS1, which links HvAP2-dependent changes in floral organ development with a specific MADS-box gene misexpression. However, they did not reveal the relationship between HvAP2 and HvMADS1 is direct or not. Our study revealed that HvAP2 can interact with HvMADS1 and such an interaction enhanced HvMADS1-induced transcriptional activity on its target gene, synergistically regulating awn development. (2) Their in situ hybridization did not detect the expression of MADS1 in awn in Bowman and *Zeol.b*. Our in situ hybridization did detect the expression of MADS1 in awn and lemma primordia at stage W4.5, 5.0 and 5.5 in Bowman (Fig. 3c), providing direct evidence to support the function of HvMADS1 in lemma and awn development. (3) Their q-RTPCR

data showed that both MvMADS1 and HvAP2 are highly expressed in developing inflorescence and developing grains, and zeo1.b showed increased HvMADS1 expression in early spikelet development (WD3.5-5.5) as compared with Bowman, while gigas1.a did not show obvious changes in HvMADS1 expression in the same period. Combined with DEGs in Zeo1.b microarray study, they assumed that HvAP2 could control floral development by modulating target gene HvMADS1. This idea is in line with ours.

Based on above mentioned comparison, we have revised corresponding part in the Discussion section as “A previous study assumed that HvAP2 could control floral development by modulating its targeting genes including HvMADS1 (Shoesmith et al., 2021). Our genetic and biochemical analyses results in this study verified that HvAP2 interacts with HvMADS1, which enhances HvMADS1-induced transcriptional activity on its target genes, thus, synergistically regulating awn and lemma development” (lines 449-453).

Therefore, although our work was an extension of Li’s to reveal underlying molecular mechanisms, the HvMADS1 alleles used in our study is different from that used by Li. In addition, Li’s work was carried out in an Australian laboratory, ours was done independently in a Chinese laboratory.

2nd Reviewer comment#2-2: Thank you for the clarification.

Answer: Thank you for your encouragement.

Comment 3: One major concern is the manuscript’s message that grain differences due to a loss of HvMADS1 function are caused by changes in lemma/ awn growth. At multiple points in the manuscript, grain changes are described as being caused by the changes in lemma development (eg. line 26 “HvMADS1 positively regulates awn length and lemma width, leading to increased grain size and weight”). However, they only show a correlation and do not provide evidence for a causative relationship, This

is also a concern with the research in rice cited in their paper. Alternative or additional possibilities should be addressed or considered, such as a role for HvMADS1 in the carpel, caryopsis or grain which could contribute to the grain differences, especially as HvMADS1 is relatively highly expressed in developing carpels and caryopses. Furthermore, the putative direct target of MADS1, HvDL clearly plays a major role in the carpel according to their manuscript (line 250-251 “hvd1 mutants were unable to set seeds due to defective carpels”), consistent with its well-known role in rice to specify the carpel (Yamaguchi et al., 2004). Thus, both HvMADS1 and HvDL could influence carpel characteristics.

Answer: This is a wonderful comment that helps us a lot to clarify the function of HvMADS1 in grain size.

To address your concerns, we have performed additional analyses, provided more results, and revised the corresponding parts. For carpel: 1) we have compared the carpel size between WT and mads1 mutant, added the results in Supplementary Fig. 3, and revised the Results part as “To rule out the effect of carpel size on grain size, we also compared carpels size between WT and mads1 mutant, and found no significant difference in the carpels size between WT and mutant at W10.0 stage (Supplementary Fig. 3a, b)” (lines 168-171); 2) We also added the carpel phenotype of dl mutant in Supplementary Fig. 7h, and revised corresponding part in the Discussion part as “Interestingly, HvDL also controls carpel properties (Supplementary Fig. 7h) and the number of lemma vascular bundles (Supplementary Fig. 10i), suggesting that there may be other proteins or pathways that control HvDL expression to regulate floral organ properties. Further investigations into upstream and downstream regulatory networks of the HvMADS1–DL module may help to identify the molecular regulators of floral organ development, including lemmas, in barley” (lines 426-432). Although HvMADS1 is highly expressed in developing carpels, the observed no changes in carpel phenotype between WT and mutant implied that the function of HvMADS1 on the carpel may be reductant with other genes, so we have also discussed the need for further analysis of higher-order mutants of the HvMADS1 gene and other barley MADS-box genes (lines 376-379).

For endosperm: we have added grain size phenotype of the cross between WT and mutant in Supplementary Fig. 3, and revised the Results part as “In addition, to rule out the effect of endosperm on grain size we crossed *mads1* (♀) with WT (♂), and found that grain size of all F2 progenies is similar to that of WT (Supplementary Fig. 3c, d) (lines 171-173).

In addition, we have discussed this issue, as suggested, in the Discussion section as “Generally, cereal grain size is mainly determined by maternal spikelet hull, pericarp, overall pistil (including carpel), and endosperm. In this study, we also demonstrated that changes in *mads1* grain weight is highly associated with changes in lemma width (Fig. 1i, 2e). We also ruled out the carpel and endosperm as the cause of the smaller *mads1* grains (Supplementary Fig. 3a, b, c, d). Thus, the 33% reduction in bulk grain weight of *mads1* grain (Fig. 2e) is thus mainly due to the narrower lemma width; nevertheless, currently we cannot completely rule out the effect of pericarp” (lines 403-410). Notably, although the expression of HvMADS1 in developing caryopses was comparable to that of early developing inflorescences (W2.0-W4.5), trends of them were different. The expression of HvMADS1 increased as inflorescence developed but decreased as grain developed (Supplementary Fig. 4a), implying likely minor roles in caryopses development.

2nd Reviewer comment #3: Thanks to the authors for this further work. The data suggesting no effect on *hvmads1* on the measured carpel parameters is helpful. The F2 seed analyses is also very insightful about possible endosperm effects. The reworded discussion addresses my concerns in the original draft.

Answer: Thank you for your recognition of the revised manuscript. We also thank you for your constructive comments.

Comment 4: They state that HvMADS1 controls awn and lemma development by directly modulating HvSHI and HvDL. Their gene-edited lines strongly support that HvDL promotes awn elongation and lemma lateral growth, and it was already known from loss of HvSHI function *lks1* alleles (Yuo et al, 2012) that HvSHI promotes awn

elongation, and this paper provides robust support that HvMADS1 directly binds to these target genes. These together are suggestive that the binding relationship is important for HvMADS1 control of awn length and lemma width. However, without genetic analyses to demonstrate how this relationship impacts the phenotypes of double mutants (eg. wrt epistasis), or transgenic rescue by overexpression of HvDL or HvSHI in the *hvmads1* mutants, we do not have an estimate of the importance of this binding (or loss thereof) to the phenotypes observed in *hvmads1*. So, I would tone this conclusion down.

Answer: We agreed with your comment and thank you so much for pointing this out. We have toned down the conclusion that HvMADS1 controls awn and lemma development by directly regulating HvSHI and HvDL.

We have revised our manuscript as suggested as shown in below.

- 1) We have revised the sentence in the Abstract “We defined two direct targets of HvMADS1 regulation, HvSHI and HvDL” to “We defined two potential direct targets of HvMADS1 regulation, HvSHI and HvDL” (lines 30-31).
- 2) We have revised the subtitle of this part “HvMADS1 directly modulates downstream HvSHI and HvDL to control awn size and lemma transversal growth” in the Results section as “HvMADS1 directly modulates the expression of potential downstream genes HvSHI and HvDL to control awn size and lemma transversal growth” (lines 244-245).
- 3) We have revised this sentence “These results indicated that HvMADS1 directly targets HvSHI and HvDL to regulate awn and/or lemma development through controlling cell proliferation” in the Results section as “These results implied that HvMADS1 directly targets HvSHI and HvDL to regulate awn and/or lemma development through controlling cell proliferation” (277-278).
- 4) We have added two sentences in the Discussion part as “Combining a series of biochemical experiments with the phenotypes of single mutants, we inferred that the binding of HvMADS1 to HvSHI and HvDL is critical for HvMADS1 in controlling awn length and lemma width. The future work should focus on the genetic analysis of double mutants (*mads1 shi* and *mads1 dl*), *mads1* mutants overexpressing HvDL and/or

HvSHI, and transgenic plants with site-directed mutated motifs of HvDL/HvSHI that bound by HvMADS1, to further strengthen the understanding of the biological significance of the HvMADS1-HvSHI/HvDL module” (lines 432-438).

2nd Reviewer comment #4: Thank you for the revisions. However, the revised subtitle text described in point 2 still states that HvMADS1 regulation of DL and SHI causes the changes in lemma and awn. My concern was not the confidence about whether HvMADS1 binds these targets or not, but whether this activity is directly responsible for the changes in lemma development in the *hvmads1* mutants. Similarly, I suggest that Point 3 still over-reaches. Consider rephrasing along the lines of: “HvMADS1 could regulate awn and lemma development through direct regulation of potential targets HvSHI and HvDL” and/or “HvMADS1 direct regulation of HvSHI and HvDL could contribute to HvMADS1-dependent regulation awn and/or lemma development through controlling cell proliferation”. The changes to the discussion text described in point 4 work very well to suggest caveats and future experiments.

Answer: Appreciated very much for careful and dedicated review.

1) For point 2, as suggested, we have revised “HvMADS1 directly modulates the expression of potential downstream genes *HvSHI* and *HvDL* expression to control awn size and lemma growth in transverse direction” to “HvMADS1 could regulate awn and lemma development through direct regulation of potential targets *HvSHI* and *HvDL*”. (lines 245-246).

2) For point 3, as suggested, we have revised “These results implied that HvMADS1 directly targets HvSHI and HvDL to regulate awn and/or lemma development through controlling cell proliferation” to “These results implied that the direct regulation of potential targets *HvSHI* and *HvDL* by HvMADS1 could contribute to HvMADS1-dependent regulation of awn and/or lemma development through controlling cell proliferation”. (lines 278-280).

Comment 5: I couldn't find detailed information about the exact gene edits for the HvMADS1, HvDL, HvSHI and HvAP2 alleles described in the paper, including the

changes within each allele, the screening, generation analysed or Cas9 presence/absence. It is also unclear if the hvmds1 alleles used in this paper are the same ones described in their previous publication (Li et al., 2021) – potentially hvmds1-5 is?

Answer: Thanks for the comment.

We have provided detailed information on the exact gene editing of the HvMADS1, HvSHI, HvDL, and HvAP2 alleles in Supplementary Fig. 1a, 6b, 7b, and 11b, respectively. We have also added a sentence in the M&M section as “Homozygous progeny plants of at least T2 or higher were used in the analyses” (line 489) to clarify the generations of these mutants used in this analysis. However, we didn't test for the presence or absence of Cas9 in these mutants.

In addition, we have clarified that the hvmds1 allele used in this paper is not the same as described in previous publication (Li et al., 2021) in the Results section as “We generated three knockout lines of HvMADS1 using CRISPR-Cas9 system, namely mads1-3, mads1-5 and mads1-8 in barley cv. Golden Promise (Supplementary Fig. 1a), with different alleles from those reported in previous studies” (lines 123-125).

2nd Reviewer comment#5: Thank you for providing all these details and clarifications. It is unfortunate that these lines were not examined for presence of the Cas9 insert. From what I see presented in Supplemental Figure 6c-e only one of the two shi alleles were characterized. Without showing that these mutants are Cas9-free, you cannot rule out the possibility that the Cas9 insert itself is causing these phenotypes and this needs to be addressed. For mads1, ap2 and dl, multiple alleles show the same phenotype, which makes contributions from the Cas9 insert to the phenotype unlikely.

Answer: Thank you very much for raising this critical issue, which helps us a lot to rule out the effects of Cas9.

We regretted that we did not screen out Cas9 initially. Objectively, multiple lines from the same gene showed similar phenotypes and they did not segregate in following

generations, which rendered us to think that the presence or absence of Cas9 unlikely has something to do with observed phenotypes.

Upon your request, we started to examine the presence and absence of Cas9 in T₂ plants of *shi-3* and *ap2-11* lines used in this study because we have only two line each for *shi* and *ap2*. We can do that because we retained the individually harvested spikes with grains and some leaves for each line.

A pair of primers for *Cas9* (Forward primer: GTA ACTCCCGTTTCGCTTGG; Reverse primer: CCGCTCTGCTTATCCCTGA) and *Actin 7* (Forward primer: AACAGAGTCACTTAATCAAATGGC; Reverse primer: GCTGGAACATTAAAGGTCTCAA) were designed, respectively. DNA was extracted from leaves retained with T₂ spikes and PCR amplification was performed to determine the presence or absence of Cas9 bands with the help of DNA agarose gel electrophoresis.

Results in Figure 1 below indicated that *Actin7* can be amplified from all plants, proving a good DNA extraction and experimental manipulation. In addition, Cas9 could not be amplified from WT but could be amplified from all T₀ plants, indicating a good negative and positive controls, respectively (Figure 1A).

Clearly, Cas9 could also been amplified from some T₂ plants of *shi-3* and *ap2-11* (Figure 1A-B).

Figure 1. Amplification of *Cas9* and *Actin 7* in different plants. (A) Amplification of *Cas9* and *Actin 7* in *shi-3* T₂ generation plants. (B) Amplification of *Cas9* and *Actin 7* in *ap2-11* T₂ generation plants.

We then compared the phenotypes of *ap2-11* and *shi-3* lines (T₂) with Cas9s with those lines without Cas9. As shown in Figure 2 in below, *ap2-11* lines with Cas9 or without Cas9 showed similar phenotypes, such as shorter awn length, smaller grain width and thickness while longer grain length as compared with WT (Figure 2A-E). Similarly, *shi-3* lines with or without Cas9 showed similar phenotypes, such as shorter awn length as compared with WT (Figure 2F-G).

Above mentioned phenotypic comparisons between lines with or without CAS9 indicated an unlikely contribution of Cas9 to the observed phenotypes in *ap2-11* and *shi-3* mutants.

Nevertheless, we will keep your concerns in our mind, and will screen only Cas-free

lines in the future studies to avoid such a problem.

Figure 2. Awn length and grain size in different mutant plants with or without *Cas9*. Phenotypes of awn length (A), grain length (B), grain width (C) and grain thickness (D) in WT, *ap2-11* without *Cas9* and *ap2-11* with *Cas9* plants. (E) Statistical data on awn length, grain length, grain width, and grain thickness in WT, *ap2-11* without *Cas9* and *ap2-11* with *Cas9* plants. (F) Phenotypes of awn length in WT, *shi-3* without *Cas9* and *shi-3* with *Cas9* plants. (G) Statistical data on awn length in WT, *shi-3* without *Cas9* and *shi-3* with *Cas9* plants.

Comment 6: More explanation of the awn removal experiment is needed. When/ how were the spikes de-awned and how long after were the photosynthetic rates determined? Also, were grain from these spikes compared with awned spikes in either genotype?

Answer: Thanks for this constructive comment.

We have added in the M&M section the explanation regarding the photosynthetic rate determination and grain weight measurement as “Barley spike photosynthetic gas exchange parameters were measured as described. Briefly, the entire awn on all spikelets of uniform main spikes of healthy WT plants at heading were cut off using

sharp blades, and the whole-spike net photosynthetic rate was measured three days after de-awning to rule out the effect of wounding. The grain weight of both WT and de-awned WT was measured at harvest time” (lines 495-500).

Additionally, we have not measured the grain weights of de-awned and non-de-awned *mads1* mutant, therefore, we have only added grain weight information of de-awned and non-de-awned WT in the Result section as “Most interestingly, the grain weight of de-awned WT was about 6% lower than that of the non-de-awned WT (Supplementary Fig. 1b)” (lines 140-141).

2nd Reviewer comment #6: Thank you for providing all these methodological details

Answer: Thank you so much for the encouragement.

Comment 7: Their conclusion that HvMADS1 promotes growth through cell proliferation is primarily based on ‘cell number’ estimates extrapolated from the measurements of cell size within a small section of the awn and lemma. They do not discuss the possibility that cell size might change throughout the organ which could also contribute to the awn length and lemma width differences – so I would use caution the interpretation to extrapolate across the entire organ and more directly state that the cell number differences were extrapolated from the measurements of cell size within a small section of the awn and lemma.

Answer: Thank you very much for raising this point. Sorry that we did not state clearly the methodology in the previous version.

For the awn, because the awn is long and thin, it is too difficult to cut the whole awn longitudinally. Therefore, we can only count the cell length by cutting the middle part of the awn longitudinally and calculating the cell number based on the entire awn length. We have revised this part in the M&M section as “For the length and number of awn cells, the awn was longitudinally cut at the middle part of the awn, cell length per awn across the entire cut region of the awn was measured, and the resulting value was divided by the awn length to yield cell numbers per awn” (lines 509-512).

We have also revised the information regarding the measurement of the cell size and

cell number in lemma cells in the M&M section as “For the size and number of lemma cells, lemma was cut transversally at the middle part of the lemma, and cell length and cell numbers were directly measured and counted, respectively” (lines 512-514). Therefore, the cell number of the lemma was not obtained by estimation.

2nd Reviewer comment #7: Thank you for the clarification that cell lengths across the entire awn were measured and how the lemma cell number and sizes were measured.

Answer: Thank you so much for the encouragement.

Comment 8: The last sentence in their discussion suggests that ‘other superior alleles of HvMADS1, natural or gene-edited, will be a key resource in the future for high-yield barley breeding’. Their previous study (Li et al., 2021) showed high levels of conservation in HvMADS1 across barley, so it is unclear how likely natural superior alleles would be found.

Answer: Thank you for this important comment. Li’s conclusion regarding high levels of conservation in HvMADS1 across barley were drawn from the coding region of HvMADS1. Therefore, we hoped to find some variations in the promoter or the first intron region of HvMADS1. Indeed, we analysed natural variations in HvMADS1 using whole-genome shotgun data from 200 domesticated and 100 wild varieties of barley5, and detected in total 559 SNPs for HvMADS1 (Figure 3a below); the population structure of these cultivars showed that they have different genetic backgrounds (Figure 3b below). However, we did not find the association of SNPs with phenotype, further research is needed to determine if there are natural superior alleles. As you suggested, we have revised this part as “so genome-edited superior alleles of MADS1 will be a key resource in the future for high-yield barley breeding.” (lines 459-461).

Figure 3. SNPs in HvMADS1 and the population structure of 300 barley varieties

2nd Reviewer comment #8: Reviewer comment: This revision works well.

Answer: Thank you so much for the encouragement.

Comment 9: Introduction incorrectly states that SHORT INTERNODES is the only gene reported in barley to control awn elongation. Shoemith et al (2021) demonstrated that loss of APETALA2 function leads to shortened lemma awns.

Answer: Thank you for pointing out the mistake.

We have revised this part in the Introduction section as “In barley, SHORT INTERNODES (SHI) controls awn elongation and pistil morphology and APETALA2 (AP2) regulates awn development and awn-lemma boundary” (lines 85-87).

2nd Reviewer comment 9: Comment addressed, thank you.

Answer: Thank you.

Comment 10: I'm not convinced that the H4 in situ shows a gradual attenuation of signal, although the awn itself is obviously smaller.

Answer: Thank you for this comment.

Although we confirmed that the expression of cell cycle-related genes is generally down-regulated in the lemma and awn of mads1 by RT-qPCR (Supplementary Fig. 5 c), to avoid confusion, we have deleted H4 related results in the revised version.

2nd Reviewer comment 10: Comment addressed, thank you.

Answer: Thank you.

Comment 11: Could the increase in Shi::LUCexpression when AP2 and MADS1 are co-expressed be explained by an additive effect? i.e. both AP2 and MADS1 binding separate reporter constructs – unless you think the proteins are provided in excess to the availability of the reporter? Something to consider.

Answer: Thanks for this insightful comment. Our current results can't rule out the possibility you raised. However, Considering the interaction between AP2 and MADS1 interactions (Fig.6a-e), this interaction can enhance the transcriptional activation of MADS1 (Fig. 6f), and genetics data (Fig. 6l), we believe that HvMADS1

synergistically regulates downstream genes with HvAP2 (lines 305-314).

Comment 12: Figure 3b far right-hand graph in panel describing the outer parenchyma cell number – was this determined from the SEM or a tissue section. This wasn't clear.

Answer: Thank you for this comment, we are sorry to have missed this information.

We have revised the legends of Fig. 3 of this part as “At W4.5, W5.5 and W6.5 stages, awns were photographed using a stereoscope and their lengths were calculated using Image J. At other stages, the awn length was measured directly with a ruler. The width and cell number of lemma were obtained by analyzing lemma sections ($n \geq 18$ individual spiklets for awn length and $n \geq 9$ individual lemma samples for lemma width and cell number)” (lines 934-938). In the M&M, we have also added detailed methodology for such analysis (Lines 509-514).

2nd Reviewer comment 11: Comment addressed, thank you.

Answer: Thank you.

Comment 12: Figure 5a and 5j – I'm not clear what the RNAseq panel is showing – does this plot represent reads from either genotype indicating a loss of reads in the mutant? Also, both panels are too small to read easily.

Answer: Thank you for this comment. We have increased the size of panel a and panel j in the revised Fig. 5.

This RNA-Seq panel indicates that numbers of reads enriched in SHI (a) or DL (j) in mads1-8 is significantly less than that in WT, proving that expression of SHI (a) or DL (j) is downregulated in mutant.

2nd Reviewer comment 12: Comment addressed, thank you.

Answer: Thank you.

Comment 13: Figure 5n 'M' from 'MADS1' is missing as an x-axis label. Furthermore, the figure caption does not describe co-expressed AP2 being part of this experiment.

Answer: Thank you very much for your comment, we apologized for the mistakes and

have corrected it in this revised manuscript as “HvMADS1 and HvMADS1-HvAP2 interaction activate expression of promoters of HvSHI (e) and HvDL (n) in barley protoplasts” (line 973-975).

2nd Reviewer comment 13: Comment addressed, thank you.

Answer: Sincerely appreciate your constructive comments again.

Response to comments of Reviewer #3

Reviewer #3 (Remarks to the Author):

The authors have addressed all my concerns appropriately. No further comments.

Answer: We sincerely appreciate your great support.